# Adenosine integrates light and sleep signalling for the regulation of circadian timing in mice

Aarti Jagannath [1✉], Norbert Varga[1], Robert Dallmann [2], Gianpaolo Rando[3], Pauline Gosselin[3], Farid Ebrahimjee[4], Lewis Taylor[1], Dragos Mosneagu [4], Jakub Stefaniak [4], Steven Walsh[1], Teele Palumaa [1], Simona Di Pretoro[1], Harshmeena Sanghani[4], Zeinab Wakaf[4], Grant C. Churchill [4], Antony Galione [4], Stuart N. Peirson[1], Detlev Boison[5], Steven A. Brown[6], Russell G. Foster[1✉] & Sridhar R. Vasudevan[4✉]

The accumulation of adenosine is strongly correlated with the need for sleep and the detection of sleep pressure is antagonised by caffeine. Caffeine also affects the circadian timing system directly and independently of sleep physiology, but how caffeine mediates these effects upon the circadian clock is unclear. Here we identify an adenosine-based regulatory mechanism that allows sleep and circadian processes to interact for the optimisation of sleep/wake timing in mice. Adenosine encodes sleep history and this signal modulates circadian entrainment by light. Pharmacological and genetic approaches demonstrate that adenosine acts upon the circadian clockwork via adenosine $A_1/A_{2A}$ receptor signalling through the activation of the $Ca^{2+}$-ERK-AP-1 and CREB/CRTC1-CRE pathways to regulate the clock genes *Per1* and *Per2*. We show that these signalling pathways converge upon and inhibit the same pathways activated by light. Thus, circadian entrainment by light is systematically modulated on a daily basis by sleep history. These findings contribute to our understanding of how adenosine integrates signalling from both light and sleep to regulate circadian timing in mice.

[1] Sleep and Circadian Neuroscience Institute (SCNi), Nuffield Department of Clinical Neurosciences, University of Oxford, OMPI-G, Oxford, UK. [2] Biomedical Sciences, Warwick Medical School, University of Warwick, Coventry, UK. [3] Department of Molecular Biology, University of Geneva, Geneva 4, Switzerland. [4] Sleep and Circadian Neuroscience Institute (SCNi), Department of Pharmacology, University of Oxford, Oxford, UK. [5] Department of Neurosurgery, Robert Wood Johnson Medical School, Rutgers University, Piscataway, NJ, USA. [6] Institute of Pharmacology and Toxicology, University of Zurich, Zurich, Switzerland. ✉email: aarti.jagannath@ndcn.ox.ac.uk; russell.foster@eye.ox.ac.uk; sridhar.vasudevan@pharm.ox.ac.uk

In mammals, the suprachiasmatic nuclei (SCN) house the "master circadian pacemaker". The SCN is comprised of multiple coupled neurones, each of which contain the molecular machinery to generate a circadian oscillation. The molecular clockwork in the SCN is aligned (entrained) to the astronomical day by changes in the quantity and quality of light at dawn and dusk, detected by specialised photoreceptors within the eye[1]. Light leads to the upregulation of two critical clock genes, *Per1* and *Per2*, whose expression sets the phase of entrainment of the molecular clockwork[2]. The SCN, in-turn, coordinates the timing of peripheral circadian clocks distributed throughout the organ systems of the body. It is worth emphasising that peripheral clocks can be entrained by multiple biological signals, including temperature, metabolites and hormones such as glucocorticoids. By contrast, the SCN is primarily entrained by light. For example, the SCN is largely unresponsive to glucocorticoids[3], although there is evidence that non-photic stimuli including exercise also contribute to SCN entrainment[4–8]. However, the SCN is sensitive to caffeine[9–11], suggesting that there exists an endogenous adenosine-based regulatory pathway that is of profound importance to circadian timekeeping. However, until now, this pathway has not been elucidated.

Caffeine is the world's most widely consumed stimulant and the most popular psychoactive drug[12]. Caffeine has multiple biological targets, but its alerting effects that counter sleep pressure[13,14] are mediated by an antagonistic action on adenosine receptors[15]. In addition to sleep, several recent studies have shown that caffeine can also affect the circadian timing system directly and independently of sleep physiology[9,10,16,17]. Caffeine has been shown to: (i) alter clock gene expression of peripheral clock cells in vitro; (ii) change the phase of locomotor rhythms in mice; and (iii) alter the frequency of action potentials within the SCN, both ex-vivo and in vivo[9,10,16,17]. In addition, caffeine enhances phase-shifting responses to light in humans[10] and also potentiates the effect of light in multiple other species[16–18].

How and why clock mechanisms should be sensitive to caffeine has remained a mystery. Because of the importance of adenosine to sleep/wake behaviour, and since sleep/wake timing and the circadian system are so intimately connected[19], we hypothesised that an adenosine-based regulatory mechanism that is sensitive to caffeine might allow sleep and circadian processes to interact in a dynamic world for the optimisation of sleep/wake timing.

In this paper, we describe a signalling pathway downstream of adenosine receptors that directly regulates entrainment and identify adenosine $A_1/A_{2A}$ receptor antagonists that specifically target this pathway. Caffeine and adenosine altered clock gene expression and circadian rhythms in vitro and in vivo via the $Ca^{2+}$-ERK-AP-1 pathway. We report that the significance of adenosine signalling is to encode sleep/wake history to the clock and modulate its response to light in mice. Furthermore, we show that clinically safe adenosine receptor antagonists delivered to mice at specific times act like light to phase-shift circadian rhythms, and also enhance re-entrainment to shifted light dark cycles. Therefore, we show that this signalling system provides a robust therapeutic target for the stabilisation of circadian rhythm disorders.

## Results

Our data, along with published findings, show that caffeine lengthens circadian period in cultured cells (Supplementary Fig. 1a, b)[9,10]. We therefore predicted that adenosine will act as a cell autonomous modulator of the circadian clockwork. Adenosine receptors occur as the $G_s$ (stimulatory) coupled $A_{2A}$ and $A_{2B}$ subtypes and the $G_i$ (inhibitory) coupled $A_1$ and $A_3$ subtypes[20]. In U2OS cells, which express both $A_{2B}$ and $A_1$ receptors

(Supplementary Fig. 1c), we observed that adenosine led to a concentration-dependent period-lengthening (Supplementary Fig. 2d, e). Further, knockdown of the $A_{2B}$ and $A_1$ receptors had opposing effects upon period length (Supplementary Fig. 1f–h), consistent with their receptor pharmacology. We also decreased adenosine by the addition of adenosine deaminase (ADA), which degrades adenosine to inosine in the culture medium and observed decreased period length (Supplementary Fig. 1i, j). Collectively, these in vitro results demonstrate a constitutive role for adenosine as a cell-autonomous regulator of circadian rhythms.

To delineate the signalling pathways downstream of adenosine receptors that couple to the circadian clock, we tested a panel of different adenosine receptor antagonists and agonists. IB-MECA, an adenosine receptor agonist, activated a classical cAMP-CREB pathway that resulted in a rapid increase in *PER1* and *PER2* expression and increased period length (Fig. 1a–e). This pathway is similarly activated in SCN neurones in response to light[1]. However, and surprisingly, we found that several adenosine antagonists also increased period length (Fig. 1f, g, Supplementary Fig. 2a–c). Importantly, these antagonists did not elevate cAMP or CRE signalling (Fig. 1h–i, Supplementary Fig. 2d–g), but they did induce the expression of *PER1* and *PER2*, suggesting the existence of an alternate pathway for adenosine-mediated regulation of the molecular clockwork. Furthermore, those drugs that targeted both $A_1$ and $A_{2A}$ receptors (CGS15943, henceforth referred to as CGS and JNJ40255392, henceforth referred to as JNJ) were the most potent class of modulators (Fig. 1j, Supplementary Fig. 2h–k). We suggest that this is due to hetero-dimerisation of adenosine receptors, which has been shown previously to differentially activate downstream signalling cascades[21]. Finally, to rule out off-target effects, we showed that the adenosine receptor antagonists failed to lengthen *Per2*-Luc rhythms after knockdown of adenosine receptors (Supplementary Fig. 2l–n).

In order to identify the CREB-independent transcriptional pathway downstream of adenosine receptor antagonism, we employed the unbiased transcription factor screening approach BC-STARPROM[22] that identifies DNA response elements (REs) using a library of barcoded reporter luciferases. CGS increased significantly the reporter luciferase signal from 8 clones within the STARPROM library (Fig. 2a). Bioinformatic analysis showed that 7 of the 8 clones contained a FOS/JUN (AP-1)-like RE, TGATTCA (AP-1 RE = TGACTCA) (Fig. 2b and Supplementary Fig. 3a). In order to experimentally identify the factors that bound to the putative RE and flanking elements in an unbiased assay, we used a biotin-tagged oligonucleotide (from clone 3) to immuno-precipitate DNA binding proteins from the nuclear fraction of U2OS cells treated with CGS for 4 h. Approximately 40 proteins were enriched by CGS (Supplementary Dataset 4). An siRNA-mediated knockdown screen of the most enriched proteins showed that *FOS* and *JUN* (AP-1) silencing decreased the expression of clone3 under baseline conditions and decreased its response to CGS (Fig. 2c and Supplementary Fig. 3b, c). Importantly, we saw attenuated period lengthening by CGS in *Per2*-Luc U2OS cells after knockdown of either or both *FOS* and *JUN* (Fig. 2d, e). Analysis of the human *PER2* gene indicated several putative AP-1 REs, including some that were highly conserved and previously validated by ChIP-Seq as reported on ENCODE[23] (Fig. 2d). By contrast, although there are putative AP-1 REs in the *PER1* promoter, they are not well conserved, suggesting that the AP-1 pathway is less important in the regulation of *PER1*. Supporting this hypothesis, adenosine receptor antagonists show greater *PER2* induction compared to *PER1* (Fig. 1j).

To define the signalling elements between adenosine receptors and AP-1, we silenced a series of members of the MAPK-ERK

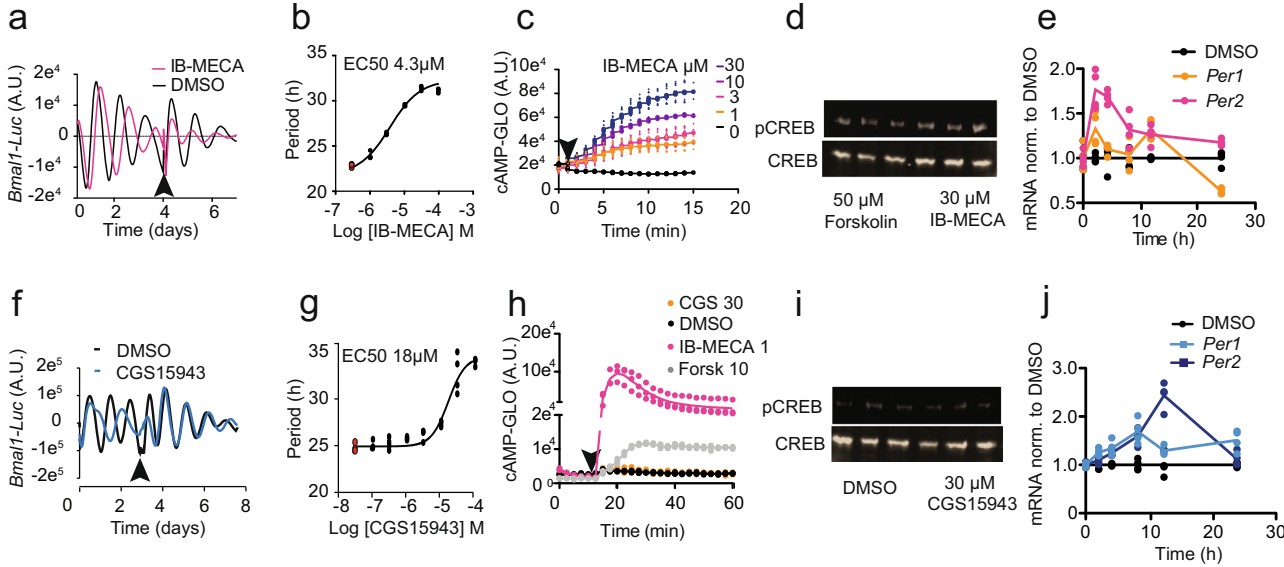

**Fig. 1 Mechanisms of action of adenosine signalling to the clock. a** 10 μM IB-MECA increased period length in *Bmal1-Luc* U2OS cells which revert to normal rhythms after washout (arrow) in a concentration-dependent manner (traces are an average of $n = 4$) as shown in **b** ($n = 4$, DMSO control samples in red); **c** IB-MECA increases cAMP levels (as monitored by the cAMP-GLO assay) in a concentration-dependent manner ($n = 6$); **d** 30 μM IB-MECA increases phosphorylation of CREB (pCREB S133) as shown by Western blot ($n = 3$), Forskolin positive control; **e** PER1 and PER2 mRNA levels in U2OS cells increase after administration of 10 μM IB-MECA ($n = 4$, $p < 0.001$ with two-way ANOVA, $p = 0.000074$ at 2 h for PER1, and $p = 0.00000000000021$ and $0.000000024$ at 2 h and 4 h for PER2, Šídák's multiple comparisons test). DMSO control in black. **f** 10 μM CGS15943 (CGS) increased period length in Bmal1-Luc U2OS cells, which revert to normal rhythms after washout (arrow), in a concentration-dependent manner as shown in **g**, $n = 4$ DMSO control samples in red. **h** cAMP increases as monitored by the cAMP-GLO assay, CGS (30 μM) administered at arrow, controls DMSO, 10 μM Forskolin and IB-MECA 1 μM shown, $n = 3$; **i** 30 μM CGS did not increase phosphorylation of CREB as shown by Western blot ($n = 3$). DMSO control shown; note **d** and **i** are part of the same blot, forskolin positive control. **j** PER1 and PER2 mRNA levels in U2OS increase after treatment with 10 μM CGS ($n = 4$, $p = 0.03$ for PER1 and $p = 0.0006$ for PER2 with two-way ANOVA). DMSO control in black.

signalling pathway that have been shown to activate AP-1[24] in U2OS cells, and then tested the efficacy of CGS at lengthening period using a concentration response curve (Fig. 2f). We then calculated EC50 of CGS at period lengthening under each knockdown condition from the curves (Fig. 2g) and found that silencing *ERK1* abolished the period-lengthening effect of CGS. By contrast, silencing the Jun Kinases (*JNKs*) had no effect upon the action of CGS (Supplementary Fig. 3b, c). Further, CGS-induced ERK phosphorylation (Fig. 2f) led to increased AP-1 activity and *cJUN* transcription (Fig. 2g). Finally, we identified $Ca^{2+}$ as the messenger activating ERK1 following CGS application (Fig. 2h). This is consistent with previous reports showing that all four adenosine receptors can modulate $Ca^{2+}$ release[20,21]. Increased cytosolic $Ca^{2+}$ can lead to the activation of ERK1/2 via multiple pathways including RAS, the protein tyrosine kinase PYK2 and Calmodulin Kinase 1[25]. In short, these results define a $Ca^{2+}$-ERK1/2-AP-1 pathway downstream of adenosine receptor antagonism in U2OS cells. These findings are outlined in Supplementary Fig. 6.

To place our in vitro findings into a physiological and behavioural context, we explored the role of adenosine signalling upon circadian rhythms within the SCN. We found that the SCN expresses both $A_1$ and $A_{2A}$ adenosine receptors in a circadian manner (Fig. 3a, b). Both CGS and JNJ lengthened period in ex vivo cultured Per2::Luc SCN slices (Fig. 3c) and adenosine itself reduced period presumably due to the prevalence of Gi coupled $A_1$ receptors. Also, CGS and JNJ increased ERK and CREB phosphorylation (Fig. 3d) and induced *Per1/2* (Fig. 3e). We speculate that this is likely due to the different receptor distribution within the SCN, with inhibitory $A_1$ receptors predominating. Here, the dominant effect of the antagonist would be inhibition of the Gi-PKA system, which would lead to an increase

in cAMP and thus CREB phosphorylation. In contrast, the $A_{2B}$ receptors present in U2OS cells are Gs and Gq coupled, therefore their antagonism would not lead to CREB phosphorylation. However, adenosine receptor signalling is known to be very complex, with adenosine receptors forming heterodimers, modulating the release of other signalling messengers and neurotransmitters, and thus engaging different pathways depending on receptor distribution[26]. Therefore, what is provided above is a proposed mechanism based on the available data.

The signalling pathways we describe for adenosine have strong parallels with those activated by light[27]. For example, *Fos* is a classical light-activated gene[28,29] and AP-1 binding to DNA increases after a light-pulse within the SCN[27]. However, although AP-1 activation is correlated with light exposure, its role in mediating the transcriptional response of the clock to light has remained unresolved. Indeed, CREB transcription alone has been considered as the primary mediator of light responses. However, unphosphorylated CREB can bind to and repress AP-1 REs[30,31]. As a result, the robust phosphorylation of CREB and the expression of FOS would act synergistically to amplify downstream AP-1 transcription. Our data would support this conclusion. A motif analysis of the genes that comprise the SCN light-regulated transcriptome[2] showed enrichment of both CRE and AP-1 REs (Fig. 2b), underscoring the importance of AP-1 in augmenting the transcriptional responses to light. These findings are also précised in Supplementary Fig. 6.

As adenosine receptor antagonists activate the signalling pathways downstream from light within the SCN, we hypothesised that these antagonists will induce phase-shifts of the circadian clock following administration at specific times during the night. In addition, several non-photic cues, including cAMP itself induce phase advances of the clock in vitro and in vivo during the

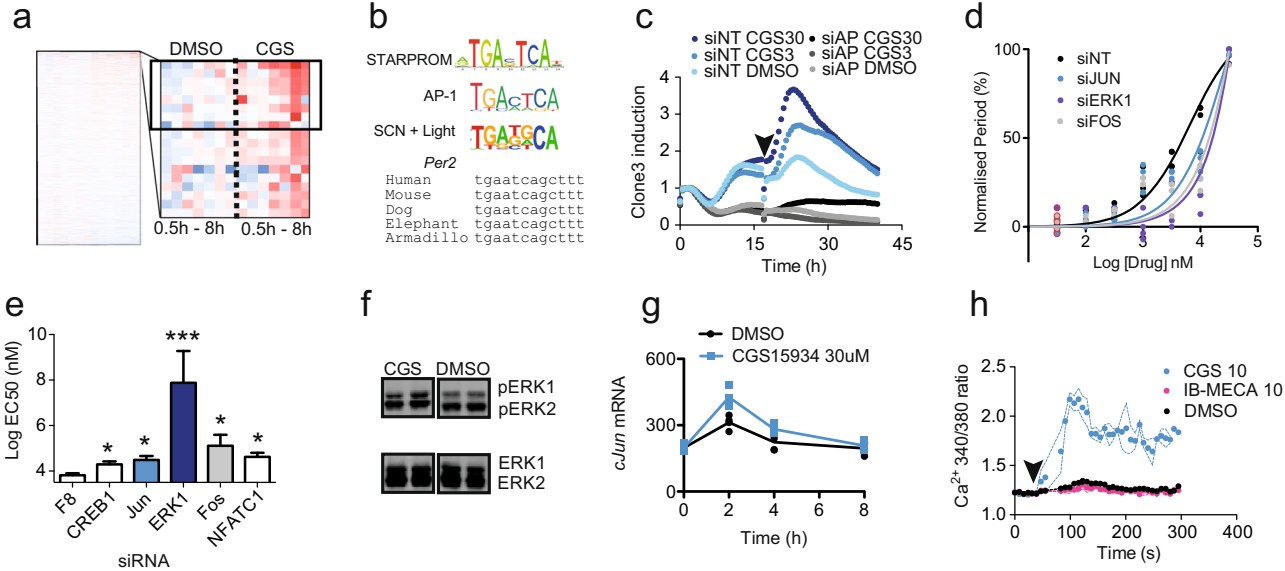

**Fig. 2 STAR_PROM identifies ERK-AP1 pathway downstream of adenosine signalling. a** Time course of RNA-seq reads for barcoded luciferase from BC-STARPROM reporter transfected U2OS cells treated with DMSO control or 30 μM CGS ($n = 2$, timeline – 0, 0.5, 1, 2, 4 and 8 h after treatment). The enlarged cluster shows the top 20 upregulated clones, of which 8 were statistically significant (boxed, $p < 0.05$, two-way ANOVA). **b** Consensus sequence from the upregulated clones with the consensus AP-1 RE and the light-regulated SCN transcriptome motif shown for comparison. The conservation of the AP-1 RE in the *PER2* gene is indicated, genomic position 2:238287740 (hg38) in humans and 1:91458384 (mm9) in mice. **c** Reporter activity of clone3 after knockdown of *FOS* and *JUN* (siAP1 – grey) when compared with a non-targeting siRNA control (siNT – blue) in response to CGS (30 μM – C30, 3 μM – C3 or DMSO, $n = 4$, single trace shown for clarity). **d**, **e** Concentration response curves and $EC_{50}$ of CGS-mediated period lengthening in Per2-Luc U2OS cells after knockdown of the genes indicated ($n = 4$, DMSO controls outlined in red * = $p < 0.05$, *** = $p < 0.001$ One-way ANOVA, Bonferroni post-hoc test); EC50 error calculated from raw data in (n). **f** Increased phosphorylation of ERK1/2 (pT202/Y204 – pERK) with 30 μM CGS treatment in U2OS cells (48% ± 18% increase, $p = 0.03$, $n = 3$ to 6, one-way ANOVA, Tukey's post-hoc test). **g** cJUN mRNA increases after treatment with 30 μM CGS – C30 ($n = 4$, ** = $p < 0.0021$ at 2 h, Šídák's multiple comparisons test). **h** $Ca^{2+}$ release in response to 10 μM CGS administered at arrow, measured by Fura2 reporter. IB-MECA caused no change. Dashed lines indicate range of data. Individual data points overlaid on all charts with line representing mean, unless otherwise indicated, error bars = S.E.M.

day[32–34] and a few enhance re-entrainment to a shifted light dark cycle[35–37]. Therefore, we tested the effects of JNJ at times that cause maximum delay in response to light (CT16), but also when non-photic cues would elicit a large advance (ZT6), and in a re-entrainment paradigm. Because JNJ has better compound solubility and a more detailed pharmacokinetic characterisation[38], our in vivo experiments were largely undertaken with this drug, although findings were confirmed with at least one other $A_{2A}/A_1$ antagonist. JNJ administered intraperitoneally (i.p.) 6 h after light onset (zeitgeber time 6; ZT6) elicited large phase advances in circadian wheel-running behaviour (Fig. 4a–c). Since adenosine-mediated regulation of the clock shares parallels with light, we also investigated the effect of JNJ administration 4 h after activity onset (the time of maximum light-induced phase delays in mice) and found that JNJ similarly elicited phase delays (Fig. 4d–f). Further, we explored whether JNJ and other $A_{2A}/A_1$ antagonists could enhance re-entrainment to a shifted light-dark cycle, and observed this to be the case (Fig. 4g–i). In agreement with our in vitro findings, drugs that targeted both the $A_{2A}$ and $A_1$ receptors were the most effective at phase-shifting and enhancing re-entrainment (Fig. 4j, k). We then studied the properties of Istradefylline (KW6002), which is an $A_{2A}$ antagonist already in clinical use for the treatment of Parkinson's disease. KW6002 significantly phase shifted circadian rhythms in wheel-running behaviour (Supplementary Fig. 4c–h, Fig. 4j). Collectively, our data show that adenosine receptor antagonists have a significant and direct effect upon circadian behavioural rhythms.

Having established how adenosine regulates circadian rhythms, we next addressed the biological relevance of this phenomenon. Extracellular adenosine levels track metabolic activity and are thus key indicators of metabolic state. Within the brain, the build-up of adenosine, presumed to be from glial and astrocyte sources, provides a strong correlate of the sleep homoeostatic drive[13,39], and these changes are monitored by different parts of the brain[40], most notably by the basal forebrain[41], but also within the hippocampus[42] and hypothalamic nuclei such as the ventrolateral preoptic nuclei (VLPO), or "sleep switch"[40]. Significantly, infusion of $A_{2A}$ agonists within the basal forebrain[43] or the hypothalamus[44] will induce sleep. Here we show that adenosine within the SCN shows day–night differences, and is lower at the end of the sleep phase (Fig. 5a) when compared to the beginning of sleep. As a result, adenosine could encode information on sleep/wake history to the molecular clockwork. Indeed, there is evidence for such an interaction. For example, increasing the homoeostatic drive for sleep through sleep deprivation changes patterns of peripheral clock gene expression in both mice and humans[45,46]. Whilst such links have been reported, the mechanisms that drive these interactions have remained obscure.

We studied the effects of adenosine on circadian behaviour in vivo, by utilising the *Adk*-Tg transgenic mouse which over-expresses human adenosine kinase (ADK)[47]. ADK phosphorylates free adenosine into adenosine monophosphate (AMP), thus overexpression of ADK enhances the metabolic clearance of adenosine. The *Adk*-Tg mouse shows reduced slow wave power in all vigilance states as defined by electroencephalography (EEG), and significantly, a reduced build-up of slow-wave activity after sleep deprivation[48]. This is consistent with reduced adenosinergic tone, which we confirm in the SCN and basal forebrain (Fig. 5 a). In the mouse SCN, we predicted that reduced adenosine would lead to less inhibition through the adenosine $A_1$ $G_i$-coupled

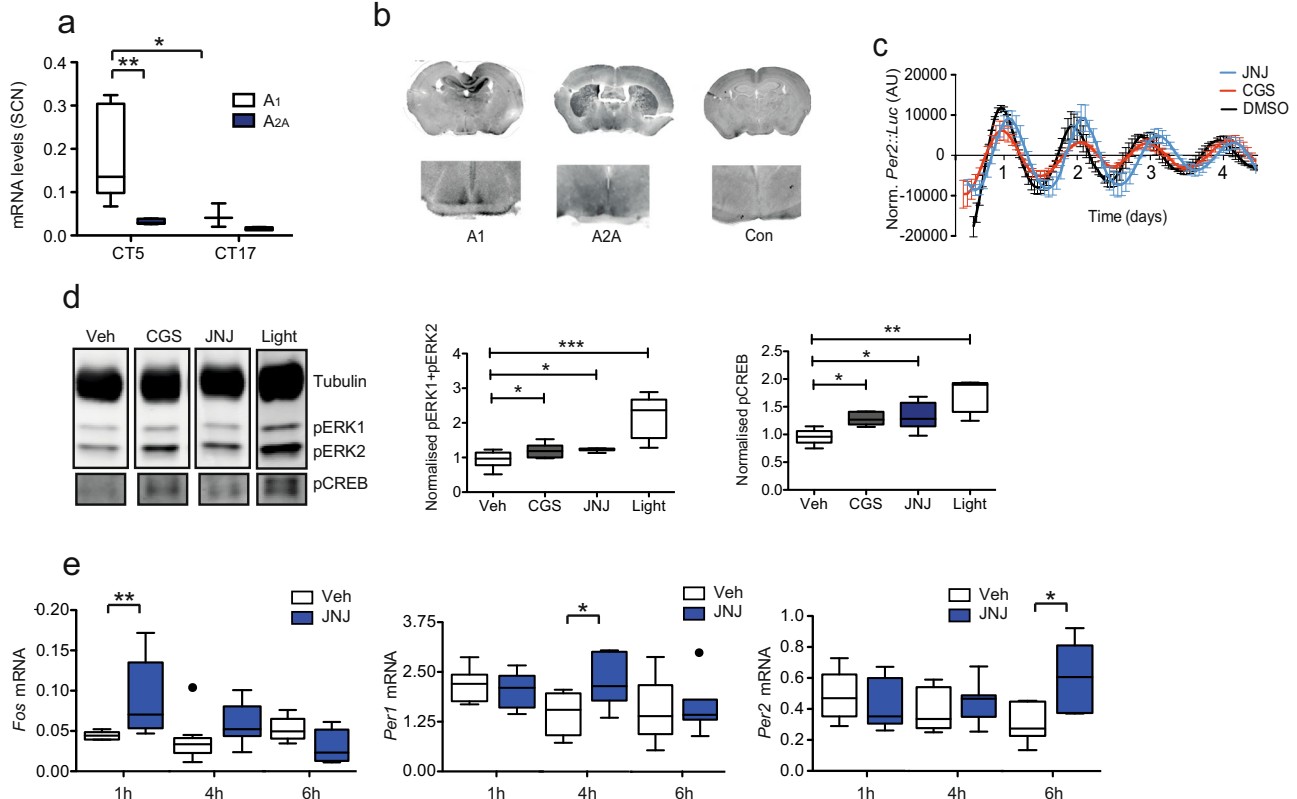

**Fig. 3 The SCN expresses adenosine receptors and responds to adenosine receptor antagonists. a** Expression of adenosine receptor (*Adora*) subtypes within the mouse SCN at two different circadian times ($n = 5$, $A_1$ at a higher concentration compared to $A_{2A}$ $p = 0.0138$, and at higher levels at CT5 (0.178 ± 0.04) compared to CT17 (0.073 ± 0.03) $p = 0.0223$, two-way ANOVA with Bonferroni post-hoc test). **b** Immunohistochemistry for $A_{2A}$ and $A_1$ receptor subtypes showing expression within the SCN (inset). **c** SCN slice cultures from *Per2::Luc* mice treated with the $A_{2A}/A_1$ antagonists CGS15943 (CGS) 10 μM (red, period length 25.1 ± 0.1 h, $n = 3$) and JNJ40255392 (JNJ) 10 μM (blue, period length 26.24 ± 0.3 h; $n = 5$; DMSO 23.1 h ± 0.3 h $n = 5$; $p = 0.0000076439$ for CGS and 0.0484 for JNJ, Tukey's multiple comparisons test. **d** Western blot with antibodies against pERK1/2 (pT202/Y204 - pERK) and pCREB (pS133) within the SCN collected from mice 45 min after i.p. injection of CGS or JNJ at 5 mg/kg at CT6, SCN collected from mice exposed to a 30 min CT16 light pulse included for comparison ($n = 7$ from two experiments of $n = 3$ and $n = 4$, uncut blots shown in Supplementary material, one-way ANOVA with Bonferroni post-hoc test). Box plots show relative expression. **e** Increases in *Fos*, *Per1*, and *Per2* mRNA within the SCN after i.p. injection of JNJ at CT6 after 1, 4 and 6 h respectively, $n = 6$–10. $p = 0.0387$ for *Fos*, 1 h; $p = 0.0183$ for *Per1*, 4 h and $p = 0.0322$ for *Per2*, 6 h, one-way ANOVA with Bonferroni post-hoc test. Tukey's box plots used throughout (central line – mean, box represents 25th to 75th percentile data, whiskers are 1.5 interquartile range).

receptor, and therefore increased period length in constant dark and larger phase shifts in response to light. Both results were observed. Greater differences in phase shifting were seen at CT22 vs CT14 ($p < 0.05$ for time with a two-way ANOVA), reflecting the naturally lower adenosinergic tone at CT14 following sleep (Fig. 5b–d). These results suggest that the biological relevance of adenosine signalling to circadian rhythms is to incorporate sleep history into the clockwork. It has been shown previously that sleep deprivation attenuates phase shifts to light, and also that A1 receptors attenuate light signalling[16,49–53]. Indeed, we confirm that sleep deprivation between CT8-14 reduces the size of phase shifting responses to a light pulse at CT14, seen but this effect is reversed when the animals are pre-treated with JNJ (Fig. 5e). To confirm that this is as a result of adenosine acting upon clock gene expression, we measured *Per1/2* expression within the SCN after sleep deprivation. We found that sleep deprivation during the second half of the light phase (ZT6-12, Fig. 5f, g) decreased *Per1/2* expression within the SCN, consistent with increased adenosine-A1 signalling. Confirmation that the mice were indeed sleep deprived comes from our demonstration that *Homer1a* and *Egr1* (both markers of sleep deprivation) were upregulated (Supplementary Fig. 5a–c[45]). Our findings are consistent with a previous study showing that there is decreased *Per2*

expression within the SCN following induced activity during the sleep phase[54], but differ from another report suggesting that there are no changes in *Per2* within the SCN after sleep deprivation[55]. We suggest that this discrepancy arises from the different methodological approaches used; both the first study and ours allowed access to a running wheel and were conducted at the same time (ZT6-12); however, the latter[55] was conducted without a running wheel, ZT0-6. Significantly, in our studies, the decrease in *Per2* in the SCN was abolished by administration of JNJ (Fig. 5h), strengthening our proposal that adenosine signals sleep–wake history to the circadian pacemaker.

Collectively, our findings suggest that light signalling to the molecular clockwork can be altered by modulating adenosine signalling. Thus, the inhibition of adenosine-A1 signalling through A1 antagonists will enhance the phase-shifting effects of light, whereas A1 agonists will diminish the impact of light. We predicted that the converse would be true of A2A signalling, and this was confirmed. Administration of specific A1 and A2A agonists and antagonists (Fig. 5i–m) altered the size of both the phase delay and advance portions of the phase response curve (PRC) to light (see below for further discussion). Thus, our results demonstrate that sleep history influences circadian entrainment through the adenosine signalling pathway.

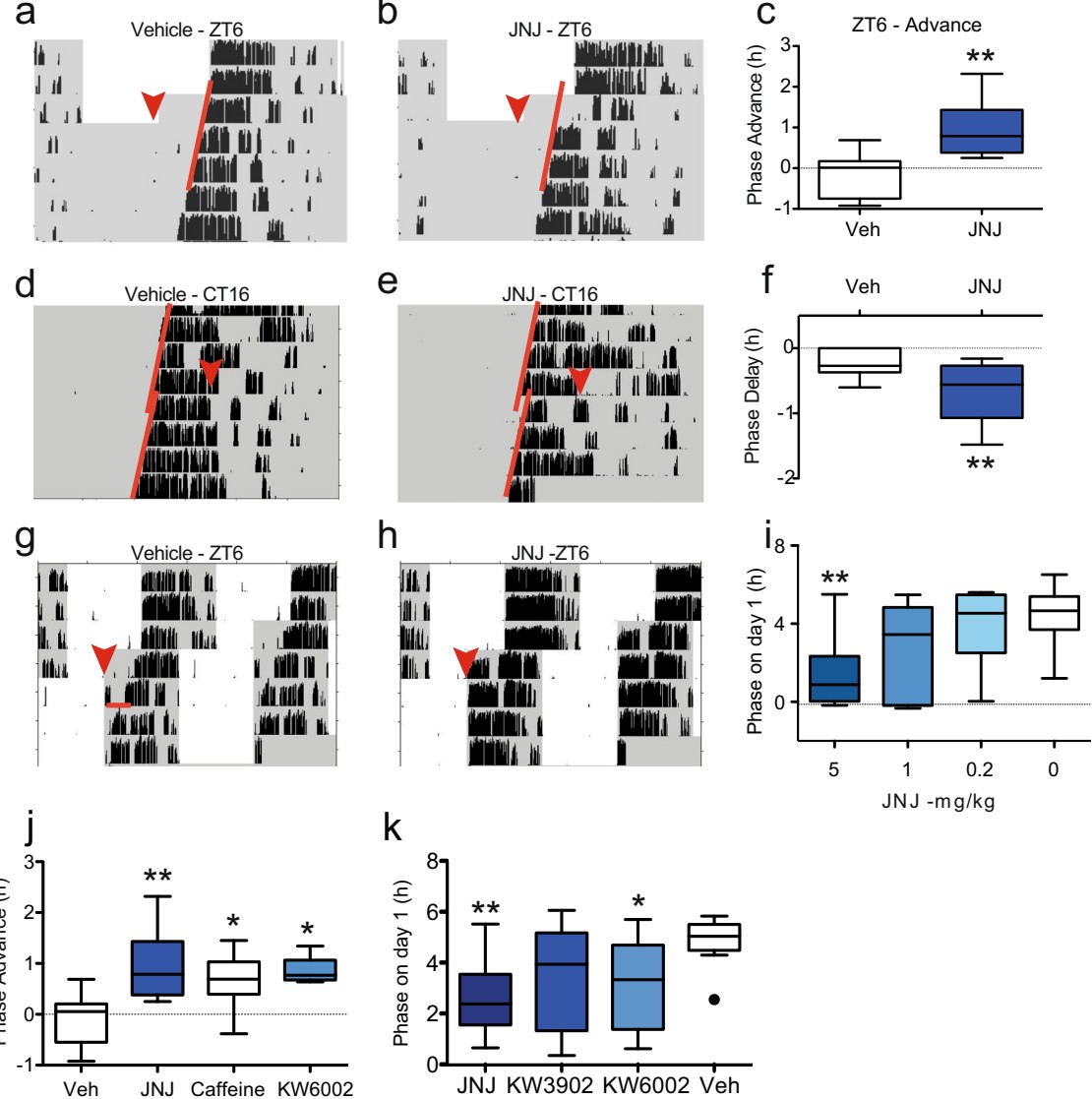

**Fig. 4 A$_{2A}$/A$_1$ adenosine receptor antagonists modify circadian behaviour. a–c** Mice were housed in 12:12 L:D cycle, (grey – dark, white – light, black vertical bars show wheel running activity). At the red arrow (ZT6), animals received an i.p. injection of **a** vehicle or **b** 5 mg/kg JNJ40255392 (JNJ) and released into constant dark (DD). Activity onset on subsequent days is indicated by red line and **c** phase shifts expressed as a Tukey's box plot ($n = 12$ and 8, ** $= p = 0.0012$, one-way ANOVA with Dunnett's multiple comparisons test). **d** Mice were housed in DD (CT16) received either vehicle or **e** 5 mg/kg JNJ at red arrow (CT16) and **f** phase shifts measured ($n = 11$ and 10, *$p = 0.007$, $t$-test). **g–i** Mice were housed in 12:12 LD received either **g** vehicle ($n = 12$) or **h** JNJ at 5 ($n = 12$), 1 ($n = 6$) or 0.2 ($n = 6$) mg/kg at ZT6 (red arrow) and LD cycle advanced by 6 h. The phase of activity was measured the day after the injection (red line) (**i**) ($n$ as above). JNJ in a dose-dependent manner enhances re-entrainment ($p = 0.0031$, one-way ANOVA). **j** JNJ at 5 mg/kg ($n = 8$) causes the largest phase shifts when administered at CT6 ($p = 0.0012$) compared with either caffeine ($n = 9$, 50 mg/kg, $p = 0.0063$) or KW6002 ($n = 5$, 1 mg/kg, $p = 0.0151$) (one-way ANOVA with Dunnett's multiple comparisons test, no significant difference between caffeine and JNJ treatment). **k** JNJ is more effective than the control and the specific A$_{2A}$ antagonist KW6002 1 mg/kg, and the specific A$_1$ antagonist KW3902 1 mg/kg for re-entrainment as measured using the 6 h phase advance protocol, $n = 12$. One-way ANOVA, Dunnett's multiple test correction, $p = 0.0023$ for JNJ and $p = 0.0342$ for KW6002 vs vehicle. Tukey's box plots used throughout (central line – mean, box represents 25$^{th}$ to 75$^{th}$ percentile data, whiskers are 1.5 interquartile range).

## Discussion

The integration of sleep/wake behaviour and circadian rhythms, via the mechanisms we outline in Supplementary Fig. 6, also provides greater clarity on the mechanisms that mediate circadian entrainment to light. Whilst previous studies have highlighted the importance of CREB-mediated transcription for circadian phase shifts to light[56], this is not the only pathway. It is important to stress that CREB targets represent only 20% of the light-regulated SCN transcriptome[2] and mice expressing a dominant negative CREB (CREB S133A) show only attenuated circadian responses

to light[57]. The work presented here shows that AP-1 provides an additional regulatory target for both light and adenosine. AP-1 binding to DNA and *Fos* induction has long been used as a robust marker of SCN activation[27–29], and *Fos* knockout mice show attenuated responses to light[58], but the signalling function of FOS has remained poorly defined. The clear relationship between FOS activation and *Per* transcription is discussed in the results and illustrated in Supplementary Fig. 6. However, we speculate that this is unlikely to be the only role for FOS. For example, FOS has recently been shown to play a critical role in initiating chromatin

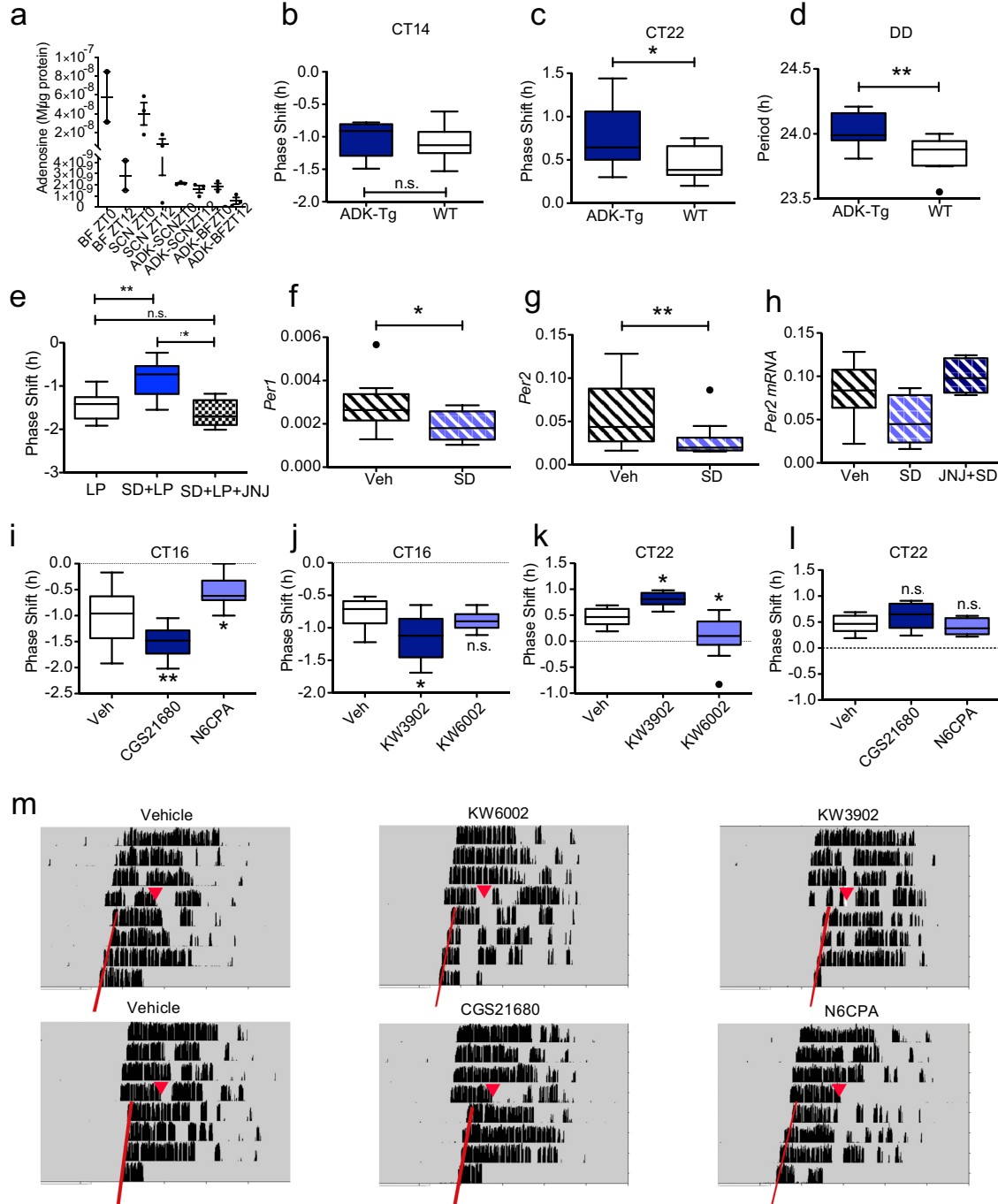

**Fig. 5 Altered levels of endogenous adenosine regulate circadian rhythms in vivo. a** Basal forebrain (BF) and SCN were collected from wild type and human adenosine kinase overexpressing transgenic mice (*Adk*-Tg - ADK) at indicated circadian times. Each sample is a pool of 5–7 individual tissue punches. **b** Phase shifting in *Adk*-Tg animals (ADK-Tg) and wild type (WT) in response to a 1 h 200 lux light pulse at CT14 (*n* = 10 and 9, no significant difference, *t*-test) (**b**) and CT22 (**c**) (*n* = 14 and 8, *p* = 0.0439, *t*-test). **d** Free running period of the same mice in DD (*n* = 12 and 9, *p* = 0.0071). * = *p* < 0.05, ** = *p* < 0.01 *t*-test. **e** Behavioural phase-shifts in response to a 30 min 20 lux light pulse (LP) at CT14 (*n* = 14) is attenuated by sleep deprivation (SD, *n* = 10) for the previous 6 h (*p* = 0.0001, *n* = 9–11), but potentiated when pretreated with 5 mg/kg JNJ (*n* = 9, *p* = 0.0007) one-way ANOVA with Tukey multiple comparisons. **f** Sleep deprivation (ZT6-12) reduces the expression of *Per1* (*n* = 11,11, *t*-test, *p* = 0.031) and **g** *Per2* mRNA (*n* = 12,12 *t*-test, *p* = 0.025) within the SCN. **h** JNJ (5 mg/kg i.p. administered at ZT6) reverses the attenuation of *Per2* within the SCN (*n* = 9,11, 8, *p* = 0.0006, one-way ANOVA with Dunnett's multiple test correction). **i–m** Behavioural phase shifts in response to light and drugs that modulate adenosine signalling via specific A$_1$ or A$_{2A}$ receptor agonists or antagonists: **i** N6CPA (A$_1$ agonist, 0.5 mg/kg, *p* = 0.0164) and CGS21680 (A$_{2A}$ agonist, 0.5 mg/kg, *p* = 0.0046) (*n* = 12, 9, 13) and **j** KW3902 (A$_1$ antagonist, 1 mg/kg, *p* = 0.0051) and KW6002 (A$_{2A}$ antagonist, 1 mg/kg) (*n* = 11, 9, 11). In **i** and **j** the phase-delaying phase shifts induced by a light pulse (30 min, 10 lux) delivered at CT16 are shown. For comparison, the phase-advance induced by a 30 min 100 lux light pulse at CT22 is shown in **k** (*n* = 10, 9, 10) and **l** (*n* = 10, 8, 8) (*p* = 0.0188 for KW3902 and *p* = 0.0078 for KW6002. One-way ANOVA for **i–l**, Dunnett's multiple test correction. Tukey's box plots used throughout (central line – mean box represents 25$^{th}$ to 75$^{th}$ percentile data, whiskers are 1.5 interquartile range). **m** Representative actograms are shown for **i–l**.

opening in response to neuronal activity within the dentate gyrus[59]. It is possible, therefore, that FOS may also induce similar changes in chromatin within SCN neurons. By changing chromatin accessibility to transcription factors, entrainment will be subject to additional regulatory controls. As illustrated in Supplementary Fig. 6, our results suggest that whilst CREB/CRTC1-CRE transcription drives a rapid increase in *Per1* transcription, AP-1 activation via FOS/JUN mediates a slower transcriptional drive upon *Per2*. Such a differential activation of *Per1/Per2* expression would explain why *Per1* rises rapidly within the SCN after a light pulse, whilst *Per2* shows a slower rise and decline in expression levels[60]. The functional significance of this differential activation remains to be fully resolved, but presumably relates to an increased level of precision and signal strength for circadian entrainment via these pathways.

Under conditions of sleep deprivation, corresponding to a high adenosinergic tone, adenosine attenuates light-induced phase shifts. This observation is interesting in the context of entrainment. In all species studied, there is a phase response curve (PRC) to light such that light delays the circadian clock at dusk and advances the clock at dawn[61,62]. In this manner, light exposure at dawn and dusk ensures that circadian rhythms are appropriately aligned/entrained to the solar day on a daily basis. Mice have circadian rhythms that are significantly shorter than 24 h, and so will show a daily phase advance in behavioural rhythms. As a result, mice require a daily phase delay to ensure that the clock is entrained to the 24 h day. Without such delaying cues, the endogenous delaying sleep/wake cycle of mice will drift earlier in time. Significantly, mice would have high levels of adenosine at dawn (following a night of activity), whilst adenosine levels will be low at dusk (following a day of sleep). Because adenosine acts to inhibit the effects of light on the circadian clock, dusk light will induce large phase delays, whilst dawn light will induce small phase advances. Whilst these observations may not hold true in all species, suggesting there are more factors at play here than just adenosine, we speculate that the reverse would be true in humans, as we sleep during the night and have longer clocks (>24 h). In this manner, light and sleep/wake history could interact to achieve optimal sleep/wake timing (Supplementary Fig. 7).

## Methods

### Experimental models

*Animals.* C57Bl/6 mice, Period2::Luciferase transgenic mice[1] and AdK-Tg[2] were used. All studies were conducted on animals over 50 days of age, male only in the case of behavioural studies, females also included for ex vivo work. Unless otherwise indicated, animals were group housed with food and water ad libitum under a 12:12 h LD cycle. All procedures were performed in accordance with the UK Home Office Animals (Scientific Procedures) Act 1986 and the University of Oxford's Policy on the Use of Animals in Scientific Research (PPL 70/6382, PPL 8092CED3), as approved by the local Animal Care and Ethical Review committee (ACER). Animals were sacrificed via Schedule 1 methods in accordance with the UK Home Office Animals (Scientific Procedures) Act 1986.

*Cell lines.* Cells from the U2OS (HTB-96) line were obtained, tested and identified. The cells were cultured in T-75 flasks, in Dulbecco's modified Eagle medium (DMEM) supplemented with 10% foetal bovine serum (FBS) and 1% penicillin-streptomycin (P/S) at 37 °C in a humidified atmosphere at 5% $CO_2$. The cells were sub cultured every 2 to 4 days in a 1:2 to 1:6 ratio, in accordance with the ATCC recommendations. The cells were counted using a haemocytometer, and their viability in a culture was maintained at above 95%, measured with a Presto Blue exclusion assay according to the manufacturer's specifications.

Two clock reporter lines were used:

1. Per2-Luc U2OS: Bioluminescence recording in phenol-red free DMEM supplemented with B-27 and 100 µM luciferin potassium salt. Cells synchronised with 100 nM dexamethasone.
2. Bmal1-Luc MMHD3: Cultured as reported in ref. [3]. Briefly, cultured in RPMI medium supplemented with 10% FBS, 1 × PSG, 10 µg/ml insulin, 55 ng/ml epidermal growth factor (EGF) and 16 ng/ml insulin like growth factor-II (IGF-II). Cells synchronised with 200 nM dexamethasone followed

by bioluminescence recording in 25 mM HEPES-buffered serum-free explant medium (pH 7.4) containing B-27 and 1 mM luciferin.

## Method details

*Animal drug administration.* The drugs were formulated in a vehicle consisting of 5% Kolliphor-HS15 (Sigma-Aldrich, UK) 5% Cyclodextran (Sigma-Aldrich, UK) in 0.9% saline. This vehicle was warmed to 37 °C and the indicated drugs kept at a 100x stock in DMSO and mixed at the appropriate concentration to be administered at approximately 10 ml/kg intraperitoneal injection ($n = 6$ to 15, indicated in each figure legend). Where the time point of injection was in the dark, the procedure was conducted under dim red light.

## Behavioural assays

*Phase shifting.* C57Bl/6 male mice (80 days or older) were maintained on running wheels in light tight chambers on a 12:12 LD cycle on stable entrainment, were released in complete darkness. At set times as indicated in the experiments, the animals received an intraperitoneal injection (10 ml/kg) of drug constituted in sterile saline with 5% Cyclodextrin (Sigma-Aldrich) and 5% Kolliphor (Sigma-Aldrich) and then allowed to free run in DD, running wheel activity data were collected and analysed on Clocklab (Version 5, Actimetrics, Wilmette, IL).

*Re-entrainment.* C57Bl/6 male mice (80 days or older) were maintained on running wheels in light tight chambers on a 12:12 LD cycle (100 lux from white LED lamps) and injected with drugs as above at ZT6. The LD cycle was immediately advanced by 6 h. Some adenosine receptor antagonists induce activity, which itself can shift the clock. Therefore, running wheels were locked for 12 h immediately following drug administration. Onset of activity on each day was used to measure phase relative to the LD cycle, data analysed on Clocklab.

*Sleep deprivation studies.* C57Bl/6 male mice at 80–120 days of age were group housed under a 12:12 LD cycle for 2 weeks with food and water ad libitum for tissue collection, or singly housed in constant dark with access to running wheel for behavioural studies. The sleep deprivation protocol consisted of novel object introduction and manual gentle handling between ZT6-12 or as listed in appropriate figure legend. If tissues were required, at the end of the protocol animals were sacrificed and tissues collected, otherwise animals were returned to the light tight chambers and activity recorded continued. Where indicated in the experiment, drugs were administered at the start of the sleep deprivation procedure as above.

*ADK-Tg studies.* 14 ADK-Tg male mice (previously described in[2,4] at 50–80 days and 10 age-matched C57Bl/6 control male animals were housed in cages with passive infrared sensors under a LD12:12 (light levels, light source). Phase shift assays were carried out at CT14 and CT22 (1 h light pulse, light levels) after a single day in DD and onset of activity on the subsequent 7 days was used to determine phase. Data were analysed on Clocklab as above.

## Tissue collection

*SCN tissue collection.* Animals were housed under a 12:12 LD cycle for 2 weeks with food and water ad libitum. Animals then received either a drug injection with alteration in the light dark cycle as described in the experiment and at set points following drug administration were sacrificed by cervical dislocation. Sham-treated mice not given a light pulse were dissected at each time point. Brains were removed and placed into a brain matrix (Kent Scientific, Torrington CT, USA). Per2-Luc transgenic animals[1] were obtained from Prof. Joseph Takahashi, Northwestern University.

*For punches for mRNA/protein analysis.* A skin graft blade (Swann-Morton, Sheffield, UK) was positioned at Bregma −0.10 mm. A second blade was placed 1 mm caudal from the first, and a 1 mm thick brain slice was dissected. SCN punches were taken using a sample corer (1 mm internal diameter, Fine Science Tools GmbH, Heidelberg, Germany) from the brain slice ($n = 4$), flash frozen on dry ice and stored at −80 °C prior to RNA extraction.

*For SCN slice culture.* Following euthanasia with isoflurane followed by cervical dislocation, the animal was enucleated and the brain was removed using standard methods. The brain was further blocked by removing the cerebellum and frontal cortex whilst preserving the integrity of the SCN and further sliced to 250 µm sections in NMDG aCSF using a compressatome (ref below). (93 mM NMDG, 2.5 mM KCl, 1.2 mM $NaH_2PO_4$, 30 mM $NaHCO_3$, 20 mM HEPES, 25 mM glucose, 2 mM thiourea, 5 mM Na-ascorbate, 3 mM Na-pyruvate, 0.5 mM $CaCl_2$ and 10 mM $MgSO_4$). The slices were further micro dissected under a microscope and transferred to membrane (Millicell Cell Culture Insert, 30 mm, hydrophilic PTFE, 0.4 µm, Millipore) in recovery media consisting Hanks Balanced Salt Solution containing 100 nM MK801, 2.5 mM AP-V and 3 mM Glutathione reduced ethyl ester for 1 h at 37 degrees and then cultured in 500 µL DMEM containing B27 in a 5% $CO_2$ incubator.

## In vitro/ex vivo experiments

*RNA extraction.* Total RNA was extracted using the microRNeasy column method (Qiagen, Hilden, Germany). Quality and quantity of RNA were measured using an Agilent Bioanalyzer and a Nanodrop1000 (Thermo Fisher Scientific, Waltham, MA USA), respectively.

*Quantitative PCR (qPCR).* RNA samples were prepared as described for microarray hybridisation. cDNA was synthesised with a qScript cDNA synthesis kit (Quanta Biosciences, Gaithersburg, MD), and quantitative PCR (qPCR) was conducted with Sybr green. Relative quantification of transcript levels was done as described previously[5]. The geometric mean of a minimum of three housekeeping genes was used for normalisation (*Gapdh, ActB, GusB* and *Rps9* for example).

Primer sequences in Supplementary dataset 1 and from ref. [6]

*RNAi.* siRNA sequences in Supplementary dataset 2 were transfected with Lipofectamine 3000 when in combination with plasmid or Lipofectamine RNAiMax when alone (manufacturer's instructions followed in both cases).

*Luciferase cell-based circadian assays.* For siRNA- and drug-based experiments, cells were seeded at 5000 per well in native medium into white 384 well plates and transfected with 50 nM siRNA the next day. Two days later, the cells were synchronised with 100 nM dexamethasone and the medium was replaced with requisite drugs formulated in DMSO as a 1000x stock (drug sources in Supplementary dataset 3), and then sealed. *Per2*-Luc rhythms were recorded from a BMG Labtech Fluostar Omega plate reader maintained at 36 °C and readings taken from each well every hour (software – Omega Version 5.10 R2). Data were then analysed using Multicycle rhythm analysis software (Actimetrics – Version 1.505).

*STAR-PROM assay.* This was conducted as previously reported in ref. [7]. Briefly, U2OS cells were cultured in 6-well plates and transfected with 1 ug of the STAR-PROM plasmid library with X-TremeGENE (Roche). After 24 h, cells were serum starved for 16 h and 30 μM CGS15943 or DMSO was added to the cells and RNA was extracted at set time points, the luciferase transcript labelled and barcoded for Illumina libraries preparation exactly as described in Gosselin et al.[22]. Library quality and size were confirmed with a bioanalyser (Agilent) and spiked with 10% of φX 174 (φX) bacteriophage DNA library to balance for AT/GC content. Libraries were loaded on a rapid-mode flow cell and sequenced inside a HiSeq 2500 machine (Illumina), which produced ~100 M single reads of 100 nt each per experiment. Bioinformatic analysis of reads was conducted as described in Gosselin et al. briefly FastQ files were uploaded onto Galaxy (https://use.galaxy.org) where 3′ end of the barcode was isolated from the luciferase sequence and demultiplexed according to the index used to label each sample. The reads were trimmed to isolate the bar code region and then counted in each sample. The counts were then normalised and fold changes calculated in R (https://github.com/randogp/STARprom).

*Mass spectrometry assay on STAR-PROM clone.* The clone3 sequence was synthesised as both forward and reverse complement strands by Sigma Aldrich, UK. The forward strand was tagged with biotin at the 3′ end and strands were annealed using New England Biolabs buffer 3. 5 million U2OS cells were treated with CGS15943 30 μM or DMSO control for 4 h followed by nuclear protein extraction using NuN buffer[8]. The nuclear extract was incubated with the biotin tagged strands and then the complexes were pulled down with Dynabeads M-280 Strepdavidin tagged magnetic beads (Invitrogen). For these steps, the protocol published by ref. [9] was followed. The pulled down proteins were then fragmented by filter assisted trypsin digestion and analysed with LC-MS/MS mass spectrometry (Q Exactive™ Hybrid Quadrupole-Orbitrap™ Mass Spectrometer). The resulting reads were sorted by degree of enrichment in the CGS-treated samples and top 40 transcription factors chosen for further analysis (Supplementary dataset 4). These were then further characterised by their expression levels in both U2OS cells and the SCN (data from NCBI GEO / BioGPS.org) and the presence of a putative binding element in the *PER2* promoter (analysed on JASPAR). The resulting top 20 were then further analysed with knockdown and reporter assays as detailed below.

*CRE-Luc and STAR-PROM reporter assays.* Clone 3 was reconstructed by ligating 6 tandem copies of the sequence GCCAACGTAATCACCCAGTGATTCATC-TAATTTCGCGAAGCGATTTTTTGTGTTTGACCTAGCATTGGCCGG-GAAGCTGCAG flanked by the sequences GAAGGCTAGCAG and GAAGCTTAG to enable insertion between the NheI and HindIII sites of the pGL4.24 plasmid (synthesised by GeneArt, Invitrogen).

CRE-Luc (pGL4.29[luc2P/CRE/Hygro] Vector, encoding Luc2P luciferase under the control of a strong cAMP response element (CRE) promoter containing 3 CRE elements within its sequence; Promega).

These constructs were transfected at 100 ng/well of a 384 well plate into U2OS cells (4000 cells per well) cultured as above with Lipofectamine 3000 following manufacturer's instructions, either on its own or in combination with 50 nM siRNA. After 24 h, the cells were maintained in serum-free DMEM containing 1 mM luciferin for 12–16h after which the drugs were added as described under libraries and screening methods. Luminescence values were measured realtime in

FLUOstar OPTIMA or OMEGA Microplate readers. For comparative experiments, drug concentrations of 10 μM were used as they were found to be optimal for keeping cell toxicity below 5%, measured with the Presto Blue exclusion assay. In the case of CRE-Luc, forskolin was used as a positive control, and 1% DMSO was used as a negative control.

*cAMP GloSensor assay.* The GloSensor™ cAMP Assay (Promega) was used for detecting changes in the intracellular levels of cAMP. Initially, a Greiner Bio-One 96 Well Plate (white, TC treated) was seeded at 8000 cells per 100 μl well, as described above. After 24 h, the cells were transfected with the pGloSensor™ cAMP Plasmid using the Lipofectamine3000 transfection reagent as above. Between 24 and 48 h later, allowing for the accumulation of the biosensor, the medium in the 96-well plate was changed to CO$_2$-independent medium supplemented with GlutaMAX™ (Life Technologies), 10% foetal bovine serum and 2% GloSensor™ cAMP Reagent. The plate was then stored for 2 h at room temperature to equilibrate. Following the incubation period, the cells were treated with relevant drugs and their luminescence values were measured after 20 min in the BMG FLUOstar OPTIMA Microplate Reader, with forskolin as a positive control, and DMSO as a negative control.

*Western blotting.* 8–10 μg of SCN or U2OS total protein in RIPA buffer were run on 4–20% SDS-PAGE gels (NuPAGE, Life Technologies), transferred using standard protocols (Bio-Rad) onto Immobilon FL PVDF membranes (Millipore), blocked with Odyssey blocking buffer (Li-Cor Biosciences, Lincoln, NE, USA) incubated with rabbit polyclonal antibodies to ERK1/2 (Rabbit anti- p44/42 MAPK ERK1/2 137F5 #4695, Cell Signalling Technology used at 1:1000), pERK (Rabbit anti- p44/42 MAPK ERK1/2 T202/204, #4370 Cell Signalling Technology, used at 1:2000), CREB (Rabbit anti- CREB 48H2 #9197 Cell Signalling Technology, used at 1:1000), pCREB (Rabbit anti- Phospho-CREB Ser133 87G3 #9198 Cell Signalling Technology, used at 1:1000) and A-Tubulin (Mouse anti- A-Tubulin DM1A #3873, Cell Signalling Technology, used at 1:5000), subsequently with donkey anti-rabbit IgG LiC or 680 secondary antibody and scanned with the Odessey Li-Cor system.

*Immunohistochemistry.* Immunostaining procedures were performed on free-floating sections. Briefly following euthanasia and dissection as outlined above, the brains were postfixed in 4% v/v para formaldehyde and 10% w/v sucrose in 0.1 M phosphate buffer saline at 4 °C overnight. The following day, brains were sliced into 50 μm sections using a compressotome and washed 3 times with 0.1% Tris-Triton solution for 10 min under gentle agitation and blocked with 5% goat serum in 0.1% Tris-Triton solution. Primary antibodies ADORA1 (AAR-006) and ADORA2a (AAR-002, Alomone Labs) were used at 1:1000 and 1:500 dilutions and incubated overnight under agitation at 4 °C. The next day following 3 washes of 15 min each with 0.05% tween20-tris solution, the slices were incubated for 1 h at room temperature with IRDYE 800 CW conjugated donkey anti-rabbit IgG (H + L) (1:5000 dilution). The slices further underwent 3 washes of 15 min each with 0.05% tween20-tris solution prior to being mounted in the presence of Prolong diamond antifade mountant. The slices were imaged using Odyssey infrared imager at a resolution of 21 μm, with 1 mm offset with highest quality.

*Ca$^{2+}$ imaging.* Compounds were tested for their ability to release Ca$^{2+}$ in U2OS cells grown in Dulbecco's modified Eagle medium containing 10% foetal bovine serum at a range of concentrations as indicated (3–4 replicates). Ca$^{2+}$ measurements were performed on confluent cells growing in 96-well plates by incubating with 2 μM fura-2-acetoxymethylester-LeakRes for 45 min in the presence of .1% Pluronic F-127 at room temperature. Fluorescence measurements were performed at $\lambda_{ex} = 340$ nm, $\lambda_{ex} = 380$ nm and $\lambda_{em} = 526$ nm in a Optima plate reader (BMG LABTECH Ltd.). Following data acquisition the $340_{ex}/526_{em}$ fluorescence (in arbitrary units) was divided $380_{ex}/526_{em}$ fluorescence and the data was expressed as a ratio.

*Adenosine measurement.* C57/BL6, housed in 12:12 LD cycle were sacrificed using cervical dislocation followed by decapitation at appropriate times. Brains were extracted and coronal brain matrices were used to obtain 1 mm sections. The SCNs or basal forebrain were punched out and rapidly transferred to a chamber containing 50 μl oxygenated NMDG media and 200 nM adenosine kinase inhibitor ABT 702 dihydrochloride (Tocris Bioscience). 5–7 punches were pooled per sample. Following 30 min incubation, the samples were spun for 10 min at 3000 RCF and the supernatant extracted. The SCNs were stored in the freezer for further protein quantification. The supernatant was filtered using a Sartorius-vivaspin 10,000 molecular weight cut-off filter to separate the media from secreted enzymes and freeze-dried for storage in −20 °C.

The assay was performed using a coupled enzyme reaction adapted from ref. [10] such that adenosine in the brain samples or standard was converted to inosine > hypoxanthine > xanthine + hydrogen peroxide using the Adenosine Assay Kit (Abcam – MET-5090). The mix was incubated for 30 min at RT in the dark in 100 mM phosphate buffer, pH 7.4. The resulting hydrogen peroxide was concurrently detected in the presence of 0.05U/sample horseradish peroxidase (Abcam) and Oxired probe at 1:25 dilution (Abcam) and the fluorescence was measured using a Berthold plate reader with excitation 535 nm and emission 587 nm. In parallel, half

the volume of the extracted samples was assayed as above, but with the omission of adenosine deaminase to account for background. The data was analysed by subtracting the fluorescence values obtained from samples from their background and normalising to protein content.

**Quantification and statistical analysis**. All statistical analyses were performed on Graph Pad Prism 8.0 software, individual tests (all two-sided) and significance levels, *n* (always biological replicates) as reported in each figure and corresponding legend. Measurements were taken from distinct samples, other than phase-shifting and re-entrainment protocols where each animal received four consecutive randomised treatments.

**Reporting summary**. Further information on research design is available in the Nature Research Reporting Summary linked to this article.

## Data availability

The datasets generated during and/or analysed during the current study are listed below, or available from the corresponding author on request. No codes were generated from the currently study that are not already publicly available. Source data as below available with this paper. RNA-seq data: https://usegalaxy.org/u/ajagannath/h/starprom2015. Mass spectrometry data: Supplementary datasets. Original uncut blots: Supplementary material. Source data for all figures are provided as a Source Data file. All other data such as PCRs and actograms are available on request from corresponding author. Source data are provided with this paper.

## Code availability

STAR-PROM analysis code: https://github.com/randogp/STARprom

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

## Acknowledgements

This work was supported by the following sources of funding: BB/N01992X/1 David Phillips fellowship from the BBSRC to AJ, WT106174/Z/14/ZMA from the Wellcome Trust to RGF, the John Fell Foundation and BB/N001664/1 from the BBSRC to SV, an N3CRs studentship to HS and NIH grants (NS103740 and NS065957) to DB. The authors would also like to thank Prof. Ueli Schibler, Geneva for critical comments on the manuscript and for providing help with the BC-STARPROM experiments supported by ERC-2009-AdG-TIMESIGNAL-250117 to Ueli Schibler. We would also like to thank Prof. John Hogenesch for *Bmal1-Luc* MMHD3 cells, Prof. Patrick Nolan for *Per2-Luc* U2OS cells and *Bmal1-Luc* plasmid and Dr. Monika Stegmann and Prof. Shabaz Mohammed for assistance with the mass spectrometry analysis.

## Author contributions

A.J., R.G.F. and S.V. conceived and designed the study. A.J., D.M., F.E., Z.W., L.T., S.W., J.S. and S.V. conducted and analysed the drug screen and validation studies including for adenosine compounds. G.C.C., A.G. and S.V. designed and oversaw the drug screen and required materials. A.J., N.V., G.R., P.G. and T.P. conducted and validated findings from STAR-PROM assay. F.E., H.S. and S.V. conducted ex vivo SCN studies including ade-nosine measurements, circadian assays and immunohistochemistry. R.D. and S.A.B. conducted studies with the *Adk*-Tg animals. D.B. provided and advised on the use of *Adk*-Tg animals. A.J., S.dP., S.N.P. and R.G.F. conducted and analysed all other in vivo studies. A.J., S.V. and R.G.F. wrote the manuscript.

## Competing interests

A.J., R.G.F. and S.V. have received funding from Circadian Therapeutics Ltd. All other authors declare no competing interests.
