## [Peer Review File · Nature Communications]

Jagannath et al., NCOMMS-19-29332A: responses to reviewers' comments in **bold**

Reviewer #1:

The reviewer states: *On several occasions the authors apply interpretations that is not in agreement with the conclusions that were drawn (and can be drawn) by the authors of the cited papers. On other occasions the interpretation of data in the context of existing data is a bit curious (the least) as articles using similar methods but acquiring different results are dismissed, whereas articles using completely different methods, but acquiring “similar” results are then described as being very similar in methods and conclusions.*

Response: This comment surfaced several times in the review provided by Reviewer 1, and these issues are addressed below.

Page 6, line 200. “Indeed there is evidence for such an interaction”. (between adenosine and the molecular clock work). This is followed by citations to papers showing changes in peripheral clock gene expression after sleep deprivation. The latter is not a very strong argument, as the same authors (the group of Paul Franken) showed that clock gene expression in the SCN does not change after sleep deprivation (Curie et al 2015).

Page 7 line 222-231. To me this is really a very strange interpretation. The authors apply a very similar sleep deprivation method (the present method is almost a copy paste of it) as Curie et al, but find a different result. The result fits better with an older study of Maywood et al, where animals were given a new running wheel, but not with the aim to sleep deprive and sleep-wake behaviour was not controlled. Then the discrepancy with the present data and the data of Curie et al is explained by a “...different methodological approach used”. As if the method used by Maywood et al is not even more different than the approach used in the present study. I think the authors should do a bit better than this. The Maywood study was not a sleep deprivation study, but an activation study aimed at increasing motor activity. The interpretation could be that the sleep deprivation method applied by the authors may have accidentally increased motor activity to levels that do change clock gene expression in the SCN. Basically this would mean that more sleep deprivation experiments, controlling for behavioural activity, need to be done to resolve this issue.

Response: The referee is primarily concerned that our results differ from those of the Franken group, which report that sleep deprivation does not alter clock gene expression in the SCN. However, the referee seems to have overlooked some important, indeed fundamental, points. Critically, the two studies were undertaken at different times. Our sleep deprivation was undertaken at ZT 6-12, whereas the Franken group (Curie et al. 2015) was undertaken at ZT 0-6. The Franken group makes the point that sleep deprivation at different times has opposite effects upon clock gene expression (Curie et al. 2013). Indeed, our unpublished data shows that sleep deprivation between ZT 0-6 similarly produces no change in clock gene expression within the SCN. Thus, in this regard, our findings are directly comparable with those of Curie et. al..

Reviewer 1 states that it is inappropriate to compare our finding with those of Maywood et. al.

We would point out that: 1) Our sleep deprivation studies, and those of Maywood, were undertaken at the same time; 2) The readout for our studies and those of Maywood used measurements of mRNA – rather than protein (Curie et al. 2015); 3) Both the Maywood study and our study used a running wheel. Collectively, therefore, it is no surprise that our results agreed with Maywood et al. and we feel fully justified in making this comparison.

Beyond our study, and the papers of Maywood et al. and Curie et al. there are no others reports of the effects of sleep deprivation on clock gene expression in the mouse SCN, and we have fully discussed our results within the context of these two studies.

Finally, in the second halve of the discussion the authors make up a story which is not substantiated with references and when searching in the literature no support for it can be found. On the contrary, an analysis has been done in this direction showing that the main proposition of the present manuscript is probably not true.

Page 8 and 9, line 267-288. In this part of the discussion (which is halve of the total discussion) the authors are overstretching the interpretation to a level that I think is unacceptable. The authors do not provide a single reference to phase response curves comparing diurnal and nocturnal phase response curves, that supports the claims made about the difference in phase shifting capacity at dusk and dawn of diurnal and nocturnal animals. However, these data do exist in the literature. In fact, an analysis done by Serge Daan, published in the Journal of Biological Rhythms (15: 195-207, figure 9) shows that what the authors suggest to be the case is probably not true. The analysis of Daan shows that there is probably no difference between the sensitivity at dusk and dawn in the phase response capacity to light between diurnal and nocturnal animals, in the way suggested by the authors.

We object strongly to the statement, “the authors make up a story which is not substantiated with references”. This is entirely incorrect and appears designed to misrepresent our work. We cite Serge Daan’s studies, which are entirely consistent with our proposal. The collection of four back-to-back papers by Daan and Pittendrigh, J. Comp. Physiol. 106, 223-331 (1976) have over two thousand citations, and collectively highlight the differences in the advance and delay portions of the phase response curves (PRCs) between nocturnal and diurnal species. Whilst a few exceptions exist, species with a shorter free-running period (generally nocturnal) show larger delays than advances, whilst species with a longer free-running period (generally diurnal) show larger advances compared to delays. This is covered in the abstract of the paper we cite (reference 43), and indeed we discussed our findings with Prof. Daan before his untimely death. We are very surprised this Reviewer has ignored these key papers and has referred instead to the transcript of a conference lecture by Serge Daan, where he discusses the “dead zone” of the PRC and not the size of advances and delays in diurnal vs nocturnal species.

Page 3, line 67: “In addition, caffeine enhances phase-shifting responses to light in humans” With reference to Burke et al 2015. This is a different interpretation of the data than Burke et al. gave in their paper, as they deducted from their data that caffeine is slowing the clock, not enhancing phase-shifting responses. It however may fit with data found in mice (Van Diepen et al Eur J Neurosci 2014;

Ruby et al J Biol Rhythms 2018) and arvicantis (Jha et al 2017 J Neurosci), and with the finding that sleep deprivation reduces phase shifting responses to light.

We have changed the text to say “caffeine elicits phase-delays in humans, and also potentiates the effect of light in multiple other species” and included the suggested references, line 69-70.

For clarification - Burke et al. show caffeine has a phase-delaying effect upon circadian melatonin rhythms in humans. He did not, as the Reviewer states, say that caffeine slows down the clock of human subjects.

Page 5 line 173. Why was JNJ given so late after activity onset (CT 16), whereas other compounds were given at CT 14?

The Reviewer is incorrect on this point. The effect of JNJ is compared with other compounds, and all were administered at CT16. We have checked the manuscript and can confirm that this is true.

Page 6 line 198: “Here we show that adenosine within the SCN shows a diurnal variation and is lowest at the end of the sleep phase...” With only two time points, this is an overinterpretation of the data. Samples should be taken at several different time points (at least 4, preferably 6 or more), evenly distributed over the course of the day, to be able to draw this conclusion. In the literature about adenosine levels in the brain it has been shown that in general (but not always) adenosine levels are low during the rest phase and high during the active phase, but the adenosine concentration differs between brain regions and the time course does not always follow the time course of sleep and waking (see for instance Zeitzer et al Sleep 2006). This is the problem with adenosine and therefore it is impossible to draw this type conclusions on the diurnal variation on just two time points.

Our experiment used a validated biochemical assay to measure adenosine. However, the assay required a large sample size for robust signal to noise detection. Thus, each data point in Figure 4a represents 5-7 pooled SCN punches, and the results presented in Figure 4a are derived from 42 animals. We have changed our statement to “adenosine levels show day-night differences, and are lower at the end of the sleep phase compared to the beginning of sleep”, line 197.

Page 6 line 211. I think a reference is missing here, probably to Palchykova et al J Neurosci 2010.

Thank you, this reference is now included (ref 35).

Page 7, line 215. “Greater differences in phase shifting were seen at CT22 vs CT14”. I agree that it really looks that way, but the statistics showing that the phase shifting capacity between wildtype and mutant is different at the two different time points is lacking. Please provide a two-way ANOVA (factors genotype and CT) showing that there is really a difference between the genotypes in this

respect.

We have undertaken this analysis and we can confirm that there is a significant difference between the genotypes (line 215).

Reviewer #2

Remarks to the Author:

In this study, the authors explored the role of adenosine on light input to the circadian timing system. This is already an area with some research history. After all, adenosine is well known to be a marker of sleep homeostasis and the levels of this chemical build in our bodies without sleep. Caffeine is one of the commonly used drugs to manipulate our alertness and it has been long known that caffeine can alter the circadian timing system through an adenosine receptor dependent mechanisms. Still there is much that we do not know including the mechanisms through which adenosine alters the molecular clock responsible for the generation of circadian oscillations and the possible involvement of the SCN.

In this study, the authors started out doing a nice series of pharmacological and genetic manipulations of U2OS cells that express both A2B and A1 cells using a Bmal-luc reporter. They delineate a pathway through which adenosine can regulate the molecular clock through the activation of Ca²⁺-EK-AP1 as well as CREB/CRTC1-cre pathways. This work is a nice bit of biochemistry but not one of interest to a general readership journal. As a result of this in vitro analysis, the authors did find a couple of pharmacological tools (JNJ and CGS) that could be used to extend their analysis to the in vivo system.

Using mice, the authors find that the JNJ compound (A1/A2A mixed agonist) causes phase shifts when administered during the day and, more importantly, causes phase delays at night. They also found that this compound could alter the rate of entrainment to a shift in the lighting conditions. They compared the JNJ compound to one dose of a melatonin agonist but did not see any effects of the drug under the conditions used. Also do they have independent measures that the melatonin drug was effective at this dose? They also found that an A2A antagonist caused phase shifts.

The authors carried out a series of experiments to better understand the function of adenosine in the SCN and circadian system. Functionally, they used a transgenic mouse model with reduced adenosinergic tone. They demonstrate that the mutant mice showed reduced slow wave power and reduced response to SD as would be expected. Did they measure adenosine? The mutant mice showed larger response to light exposure as would be predicted by the model.

They found that SD reduced light-induced phase shifts as well as Per2 expression in the SCN. They did not use the EEG to confirm SD but they did show that Homer1a and Egr1 were upregulated as expected with SD. The decrease in Per2 in the SCN was abolished by the administration of JNJ. Were the light induced phase shifts similarly restored to control levels?

Overall this is an impressive study. They started with cells in culture worked out a new pathway and

then successfully tested this pathway in vivo. There were some issues that need to be addressed.

We thank this reviewer for the very positive and constructive feedback that recognises the importance of the research conducted. We address each of the questions/comments below.

I was not convinced by the argument that their drug is better than Tasimelteon. We do not have any dose response information or independent measures that the drug is in the system. Under the conditions tested, the drug did not have a behavioral effect but we do not have the information to make a comparison. I would recommend that this be removed.

We have removed these data from the manuscript as we agree they are ancillary to our primary findings. For clarification to the reviewer, we have used a dose of the drug comparable to that used in the published Tasimelteon advisory committee meeting briefing materials by Vanda Pharmaceuticals, but are nevertheless aware that differences in routes of administration etc. may contribute to the lack of the effect seen.

For the work with the ADK overexpressing line, I would like to see that the adenosine levels were indeed reduced in the SCN. As they have a working assay, I am surprised that this measurement was not carried out.

As we did not have access to these transgenic animals at the time at which these reviews were received, we had to source these animals from Prof. Boison, rederive the strain and conduct this experiment, all of which has taken over 9 months. The results are in line with what we expect and shown in Fig 4a, line 211. We thank the reviewer for strengthening our study with this suggestion.

The authors emphasize that the administration of JNJ rescued the impact of SD on Per2, but what about on the impact of SD on behavior? Can JNJ overcome the reduction in light-induced phase shift seen after SD?

This is an important point, and we now have data to show that JNJ administration does indeed rescue the impact of SD on behaviour, this has been included in the manuscript (line 220 and Fig. 4e).

Finally, I believe that it would help the readers to explain how the SD evoked changes in the light-entrainment pathway are adaptive to a diurnal and nocturnal organism. It is briefly mentioned in the discussion but I think that this point deserves to be emphasized. Perhaps in a figure as the authors did for the signaling pathway. How is an adenosine evoked decrease in the light-input to the circadian system adaptive or functionally useful?

This an excellent suggestion and we have provided a summary figure as Supplementary Figure 7, line 286.

Figure 2 may need some work. The sections through the brain are rather beaten up. Are alternatives available? In Fig2D, I am confused about the comparisons with light. The legends state that the drugs

were administered at ZT 6, but were the mice exposed to light 6? At ZT 6, light should not have any impact.

We have replaced the image of the damaged slice (Fig. 2b) and included a control immunostained slice image. In 2d, the light pulsed samples were collected at CT16, not CT6, we apologize for the oversight and this is now corrected, line 370.

In Fig 3, please remind readers of the drugs in the legend. Also, in Fig. 3J, it looks like caffeine was equally effective as JNJ in evoking phase shifts. The readers may be interested in whether any of the findings suggest drugs like JNJ perform better than caffeine itself.

The drug names are now spelled out in full in the legend. There is no significant difference between the size of the phase-shifts elicited by caffeine and JNJ, although caffeine was administered at ten times the dose of JNJ, this information is now included in the legend, lines 391.

But again, this is an excellent study.

We thank the reviewer for their support.

Reviewer #3:

Remarks to the Author:

This manuscript addresses important questions related to how adenosine accumulation during sleep modulates the circadian system. The authors show how adenosine receptor agonists and antagonists change circadian period, intracellular signaling (e.g. induction of cAMP, pCREB, cJun, pERK, Per1 and Per2 transcripts) in U2OS cells, the SCN in vitro and in vivo, and locomotor behavior in constant darkness and a shifted light cycle. For example, they provide compelling evidence that CGS-15943 (a non-selective adenosine receptor antagonist) acts through a Ca²⁺-ERK1/2-AP1 pathway to induce Per2 and, to a lesser extent, Per1 expression, and lengthen circadian period in U2OS cells. The authors conclude that light induced phase shifts depend on sleep-induced adenosine effects on circadian period and sensitivity to light via A1 and/or A2 receptors, cAMP and Ca²⁺ and argue that blocking adenosine receptors shows promise to reduce jet lag. The authors are applauded for the broad combination of approaches (pharmacology, RNAi, luciferase reporters, locomotor behavior etc.) and the results will be interest to neuroscientists, pharmacologists, chronobiologists, and, potentially, sleep and circadian clinicians. The manuscript will be strengthened when the authors address the following major and minor concerns.

We thank this reviewer for their constructive feedback and of their recognition of the importance and relevance of our studies.

Major.

Physiological relevance. The authors argue that adenosine accumulation during sleep and loss during wake modulates circadian period and sensitivity to light. Much of the interpretation depends on

results in U2OS cells which they find express A1, but not A2, receptors.

The referee has misunderstood, we show that U2OS cells express greater levels of A2B receptor compared to A1 (Supplementary Fig. 2c) and this is in contrast to the SCN, which expressed more A1 (Fig. 2a).

They find that high doses of adenosine (100uM) lengthened period by ~1h (to 23h) in U2OS cells. It would greatly help the paper if they test this on the SCN in vitro or in vivo. It is concerning that the diverse drugs tested (an agonist, IB-MECA, and 5 antagonists with differing EC50s and specificities) all lengthen period in U2OS cells and with much greater potency than adenosine (to periods of 26-35 h).

Adenosine is a natural metabolite that is processed and degraded within seconds. By contrast, the drugs we have used are designed to have much longer half-lives and greater potency on adenosine receptors. For example, the EC50 of adenosine (in a cAMP release assay) at the A2A receptor is a 100 fold greater than for JNJ40255293, therefore it is expected that these synthetic drugs are more effective than adenosine at lengthening period. With regard to testing adenosine on the SCN, we thank the reviewer for this suggestion. We have now done this as in included as part of the manuscript (see Supp. Fig. 4, line 151)

Can the authors relate the 1-100 nM adenosine/ug protein they found in the brain to changes in period sensitivity to light? To be more convincing, we need to see adenosine and the drugs change circadian period in the SCN in vitro and, ideally, in vivo. Is there any evidence in the literature that sleep-deprivation induced changes in adenosine alter circadian timing?

Data on the effects of adenosine on SCN ex vivo have been included (Supplementary Fig. 4a-b) as suggested. There is no direct evidence of sleep-deprivation induced changes in adenosine altering circadian timing, but this relationship can be inferred from the same papers as those listed by this reviewer; “administration of a selective A1 adenosine receptor subtype agonist reduced the size of the phase shift induced by light similar to sleep deprivation (Watanabe et al., 1996; Elliott et al., 2001; Sigworth & Rea, 2003)”. Our work provides the first mechanistic evidence in this regard.

The model is intriguing, but confusing and not well supported by the data. The model (Supp Fig 6) argues that adenosine has many effects on the SCN including shifting circadian phase (when light has little effect, e.g. CT6), slowing the rate of photic entrainment (by antagonizing light effects), decreasing sensitivity to light at all times while advancing or delaying at specific times and shortening circadian period through an unstated mechanism. How do these observations explain that “adenosine integrates light and sleep signaling for the regulation of circadian timing”?

The model is based upon our findings that show, as the Reviewer states, “that adenosine slows the rate of photic entrainment (by antagonizing light effects), and decreasing sensitivity to light at all times”. As we have worked out the signalling pathways based on regulation of cAMP and Ca²⁺ influences on clock gene expression, we feel that our model represents a good approximation of the probable mechanism. These pathways are synthesised in supplementary figures 6 and 7, and the source data for each element of the pathway are indicated in the

legend. Further, we envisage that the addition of Suppl. Fig. 7 clarifies this point.

The authors are applauded for their ambitious attempts to integrate results from immortalized cells treated with drugs in vitro and mice treated with drugs delivered IP, but the data do not support the model. For example, 1) some adenosine receptor antagonists (CGS and JNJ) increase CREB phosphorylation in the SCN, but not U2OS cells,

This we reason is due to different receptor expression patterns, there are far more A2B (excitatory) than A1 (inhibitory) receptors in U2OS cells, but the converse is true in the SCN, where adenosine antagonists would be predicted to have this effect and have altered the text to clarify this (line 152). Hence, we show that the same drug will have different effects on CREB phosphorylation in U2OS cells vs SCN neurones.

2) adenosine or an agonist (e.g. IB-MECA) critical to the model weren't tested on the SCN,

Adenosine's effect on the SCN is now included, Suppl. Fig. 4a-b, thank you for this valuable suggestion.

3) similar antagonists differ in their effects on clock gene expression (eg. CGS and JNJ) in U2OS cells, and 4) similar antagonists either sped entrainment (JNJ), had no effect (KW6002, KW3092) or were not tested, although the conclusion was based on the one that sped entrainment.

These drugs are not similar in their affinities and EC50s for the different adenosine receptors (as acknowledged by the reviewer above). For example, JNJ has a much higher affinity for A1 receptors compared to KW6002, and thus its effect on entrainment on a system in which A1 receptors dominate, such the SCN, would as shown, be predicted to be much greater. A table detailing drug affinity has been included (Supplementary table 5).

The authors discuss how having different ratios of A1 and A2 receptors could make cell types respond differently to adenosine. Ultimately, we lack evidence that the ratio of A1 and A2 receptors explains how the SCN responds to physiological adenosine.

This is a good point and has been addressed in the present study using pharmacology (Figure 4). However, future work could address this issue using receptor knockout animals.

Novelty and contrasts with prior studies. Prior studies, some not cited, have concluded that adenosine modulates circadian entrainment and shown the same pharmacological effects of the drugs reported here. As summarized in van Diepen et al. (2014 EJM): "These findings suggest a role for adenosine in modulating the effect of sleep deprivation on the phase-shifting capacity of the circadian clock in response to light."

We have placed this prediction into a functional context. Our work has defined the mechanisms and the physiological relevance whereby adenosine signalling interacts with light for circadian regulation, moving the field forward by turning speculation into an evidence-based understanding.

Figure 1 has largely been done previously. For example, Isojima et al (PNAS 2009) previously showed that CGS, IB-MECA and a few other adenosine receptor agonists changed circadian period in U2OS cells.

This high-throughput screen by Isojima et al (PNAS 2009) lists these drugs in a table with no annotation or mechanistic context, alongside many other compounds that were effective at changing circadian period.

Figure 2 contradicts prior studies which also reported evidence for adenosine A1 and A2 receptors in the SCN (Chen & van den Pol, 1997), but failed to find daily rhythms in expression (Panda et al 2002, Cell. Chiang et al, 2014, PLoS Genetics).

The papers that failed to find daily rhythms were omics approaches designed to probe the entire protein/mRNA landscape. PER1/2, core clock proteins, shown by dozens of previous studies to be rhythmic in the SCN are also not reported as rhythmic in the Chiang et al. study. Therefore, these studies cannot be used as evidence for the absence of a rhythm.

Fig. 3 uses a new drug (JNJ) to show what has been reported with other adenosine receptor modulators. For example, administration of a selective A1 adenosine receptor subtype agonist reduced the size of the phase shift induced by light similar to sleep deprivation (Watanabe et al., 1996; Elliott et al., 2001; Sigworth & Rea, 2003). This effect was blocked by administration of an A1 adenosine receptor antagonist (Elliott et al., 2001; Sigworth & Rea, 2003). Administration of an A1 adenosine receptor agonist decreased light-induced expression of c-Fos (Watanabe et al., 1996).

We agree that these studies support the data presented in Figure 3. In addition, there are multiple studies that have addressed the role of adenosine in sleep regulation. However, we make the strong point that this literature is fragmentary with regard to circadian regulation and can only be placed into a biological context in the light of the novel mechanistic work we present in our manuscript.

Appropriate controls. I could not find the DMSO controls for the dose-dependent effects of the tested drugs (e.g. Figs. 1b, 1g, 1n, SF1b and SF1e) or for the immunohistochemistry on A1 and A2A receptors (e.g. Fig. 2b). Did the authors test for diurnal or circadian rhythms in A1 or A2A protein in the SCN?

The lowest concentration of the drug used in each dose response curve is 0, in other words this point is the DMSO control. As it is impossible to plot log0, we have assigned them the lowest value on the scale. These data points are highlighted in a different colour (red) as the DMSO controls to add clarity, we thank the reviewer for the important point. Negative controls for the immunohistochemistry are included in Fig. 2b and we tested for diurnal expression of receptor mRNA only, not protein. The immunohistochemistry was conducted on slices obtained from animals sacrificed in the middle of the day, when gene expression is at its highest. We did not attempt to quantify rhythmic protein expression as these receptors are difficult to analyse with western blots (as are many membrane proteins).

Did the authors omit the Methods section? For example, the paper depends heavily on drugs such as JNJ which I could not find in the literature. Can they provide more information about the full drug name or structure? It will be important for readers to know that CGS is not a universally accepted antagonist of adenosine receptors (e.g. it binds to human A1, A2, or A3 subtypes but not to rat A3; Ghai G, et al. 1987; Kim YC et al 1996).

A methods section was included in the submission. JNJ's full name is listed on line 106 in the main body of the text and also within Suppl. Fig. 2, where it is first used. We mention CGS as an A1/A2 antagonist (see line 105), as A3 is not expressed in either the U2OS system or the SCN. We have now included a table detailing the full names of the drugs used and their receptor affinities (Supplementary Table 5).

Minor.

Fig 3J text describes, but does not plot, results from IB-MECA. Fig 3c, f and j. To avoid confusion, it would help to label the y-axes as "phase advance" or "phase delay". It would help if the authors keep the same nomenclature for their drugs throughout (e.g. CGS is also called C30; IB-MECA is also called i1).

Both these suggestions have been incorporated and the erroneous reference to IB-MECA has been removed.

The authors could be less vague with the hypothesis they aim to test: "an adenosine-based regulatory mechanism that is sensitive to caffeine might allow sleep and circadian processes to interact..."

This statement comes from the end of the Introduction and now reads "an adenosine-based regulatory mechanism that is sensitive to caffeine might allow sleep and circadian processes to interact in a dynamic world for the optimisation of sleep/wake timing", line 74. We return to this point at the end of the discussion, where we detail how sleep/wake status via levels of adenosine, acts to alter light entrainment at dawn and dusk to achieve entrainment for both nocturnal and diurnal species. If this is not sufficiently clear, we would be happy to modify further.

The authors should moderate their main conclusion that they found "a novel pathway downstream of adenosine receptors that directly regulates entrainment." The pathway they propose does not seem to include novel elements and was not shown to directly regulate entrainment.

We would like to stress that there are no published studies which have described how adenosine acts upon the pathways that regulate the molecular clockwork (Ca, cAMP >> ERK >> AP-1 >> PER).

Reviewers' Comments:

Reviewer #1:

Remarks to the Author:

The authors have addressed all our points to our great satisfaction. There are two remaining points that can easily be addressed in the text:

Response: The referee is primarily concerned that our results differ from those of the Franken group, which report that sleep deprivation does not alter clock gene expression in the SCN. However, the referee seems to have overlooked some important, indeed fundamental, points. Critically, the two studies were undertaken at different times. Our sleep deprivation was undertaken at ZT 6-12, whereas the Franken group (Curie et al. 2015) was undertaken at ZT 0- 6. The Franken group makes the point that sleep deprivation at different times has opposite effects upon clock gene expression (Curie et al. 2013). Indeed, our unpublished data shows that sleep deprivation between ZT 0-6 similarly produces no change in clock gene expression within the SCN. Thus, in this regard, our findings are directly comparable with those of Curie et. al..

Reviewer 1 states that it is inappropriate to compare our finding with those of Maywood et. al. We would point out that: 1) Our sleep deprivation studies, and those of Maywood, were undertaken at the same time; 2) The readout for our studies and those of Maywood used measurements of mRNA – rather than protein (Curie et al. 2015); 3) Both the Maywood study and our study used a running wheel. Collectively, therefore, it is no surprise that our results agreed with Maywood et al. and we feel fully justified in making this comparison. Beyond our study, and the papers of Maywood et al. and Curie et al. there are no others reports of the effects of sleep deprivation on clock gene expression in the mouse SCN, and we have fully discussed our results within the context of these two studies.

1. >>The method of sleep deprivation used in this study was the one applied by the Franken group (the so-called 'gentle handling' method) and not by placing the animals in a novel wheel and increasing activity. In this context I would suggest that the authors do not simply write '... that this discrepancy arises from different methodological approaches used' on page 7 line 230-231, as this is cause for much confusion, but specify what part of the methods the authors think is important in causing the discrepancies.

We object strongly to the statement, "the authors make up a story which is not substantiated with references". This is entirely incorrect and appears designed to misrepresent our work. We cite Serge Daan's studies, which are entirely consistent with our proposal. The collection of four back-to-back papers by Daan and Pittendrigh, *J. Comp. Physiol.* 106, 223-331 (1976) have over two thousand citations, and collectively highlight the differences in the advance and delay portions of the phase response curves (PRCs) between nocturnal and diurnal species. Whilst a few exceptions exist, species with a shorter free-running period (generally nocturnal) show larger delays than advances, whilst species with a longer free-running period (generally diurnal) show larger advances compared to delays. This is covered in the abstract of the paper we cite (reference 43), and indeed we discussed our findings with Prof. Daan before his untimely death. We are very surprised this Reviewer has ignored these key papers and has referred instead to the transcript of a conference lecture by Serge Daan, where he discusses the "dead zone" of the PRC and not the size of advances and delays in diurnal vs nocturnal species.

2. >>Citation 43 (Daan and Pittendrigh 1976) is not about comparing phase response curves between diurnal and nocturnal mammals. What Daan and Pittendrigh show is that there is a large variability in phase response curves in nocturnal rodents and that there seems to be a relation with the corresponding free running period. However, when one looks at the two extreme species in this paper (Figure 3 of Daan and Pittendrigh 1976), the mouse (*Mus musculus*) and the golden (or Syrian) hamster (*Mesocricetus auratus*) it is clear that the suggestion made by the authors in the end of the discussion cannot be right. The mouse phase response curve indeed fits very well with the authors idea as it has a small phase advance part and a large phase delay part, which fits

with the idea that adenosine accumulating during the active phase is reducing the light response in the phase advance part. However, for the hamster the phase response curve shows quite the opposite phase. Both are nocturnal mammals and therefore active in the dark phase, yet in the hamster the phase advance part at the end of the night is much larger than the phase delay part at the beginning of the night. So, the statement made by the authors in their response that the data presented in Daan and Pittendrigh 1976 is 'entirely consistent with our proposal' seems to me to be incorrect.

What can be seen in Figure 9 of the Daan 2000 paper is that the average phase response curves do not differ very much between diurnal (7 species) and nocturnal mammals (12 species). In addition, it also shows that the advance and delay part of the phase response curve within the diurnal and nocturnal group do not differ that much either. Both speak against the suggestion made by the authors in the final part of the discussion in the present manuscript. I therefore urge the authors to remove this part of the discussion as it is not supported by data in the literature.

Reviewer #2:

Remarks to the Author:

This is a revised version of a manuscript titled: "Adenosine integrates light and sleep signalling for the regulation of circadian timing".

I raised a number of concerns with the previous version and feel that the authors addressed these concerns.

In broad strokes, the authors use pharmacological and genetic approaches to provide evidence that demonstrate that adenosine acts upon the circadian clockwork via adenosine A1/A2A receptor signalling through the activation of the Ca²⁺-ERK-AP-1 and CREB/CRTC1-CRE pathways to regulate the clock genes Per1 and Per2. They show that these signalling pathways converge upon and inhibit the same pathways activated by light. Thus, circadian entrainment by light can be modulated by sleep history and provides a nice concrete example of how process S and C can interact.

This is exciting and should generate broad interest. I continue to be impressed with the combination of data from cell culture, SCN biochemistry, and behaviour.

From my point of view, the problem that the other reviewers and I struggle with is that there are so many details that the main point gets lost. But the writing is generally clear and help the reader follow the linear flow of the story. The manuscript could likely benefit from even more pruning of the figures. But I do recognize that a lot of work went into this study.

I have a few additional points (minor).

In Fig1d, are negative controls available? Similarly, for Fig.1i, are positive controls available? The activation and inhibition of pCREB would be more striking with the controls.

For Fig1e, j were two way ANOVA used to assess which gene expression values are significantly different?.

For Fig1o, can the authors provide more explanation of the siRNA experiment. I was struggling with the visual display of this experiment even if the point is clear.

In Fig2a, it looks like the receptor is not expressed in the night. Can the authors provide more explanation for how they seen regulation of the light during the night? Is the receptors half-life sufficient that the A1 made in the day would be present at night?

In Fig2d the authors are showing the drugs increase pCREB. Can the authors help the reader understand how this fits with Fig1i which apparently does not show pCREB activation?

Reviewer #4:

Remarks to the Author:

This is an interesting set of experiments, pursuing a topic that was investigated by several groups about 20 years ago. The study confirms findings from prior in vivo behavioral pharmacology studies and extends these to define intracellular signaling pathways. I agree with previous reviewer comments that the results are confusing in some respects (see below). There are also issues with scholarship, noted in the first round of reviews, but not addressed in this revision. The authors fail to acknowledge prior work, although they do appear to be aware of it, based on the response to the reviews.

Introduction

1. page 2, line 56 - The authors state that "By contrast, the SCN is almost exclusively entrained by light and is unresponsive to such stimuli³". This is factually incorrect; the SCN is neither exclusively entrained by light, nor is it unresponsive to non-photic stimuli. There is copious evidence that the SCN pacemaker in sighted and blind animals can be shifted and entrained by non-photic stimuli, including scheduled activity and feeding. Sensitivity may vary by species and strain (SCN controlled free-running rhythms in some mouse strains entrain to feeding schedules; in others this is less likely, and in some species even less so; species differences in phase shifts to scheduled exercise are also known), but these effects are not trivial, can compete with light, may be more important in natural environments (more studies needed), and should not be dismissed. There are many review papers that can be cited here [below is a selection from a quick pubmed search]. This needs to be corrected so that readers new or peripheral to the field are properly informed.

- a. Webb IC, Antle MC, Mistlberger RE. Regulation of circadian rhythms in mammals by behavioral arousal. *Behav Neurosci*. 2014 Jun;128(3):304-25.
- b. Challet E. Minireview: Entrainment of the suprachiasmatic clockwork in diurnal and nocturnal mammals. *Endocrinology*. 2007 Dec;148(12):5648-55.
- c. Mistlberger RE, Skene DJ. Nonphotic entrainment in humans? *J Biol Rhythms*. 2005 Aug;20(4):339-52.
- d. Yannielli P, Harrington ME. Let there be "more" light: enhancement of light actions on the circadian system through non-photic pathways. *Prog Neurobiol*. 2004 Sep;74(1):59-76.
- e. Mistlberger RE, Skene DJ. Social influences on mammalian circadian rhythms: animal and human studies. *Biol Rev Camb Philos Soc*. 2004 Aug;79(3):533-56.
- f. Challet E, Pévet P. Interactions between photic and nonphotic stimuli to synchronize the master circadian clock in mammals. *Front Biosci*. 2003 May 1;8:s246-57.
- g. Hastings MH, Duffield GE, Smith EJ, Maywood ES, Ebling FJ. Entrainment of the circadian system of mammals by nonphotic cues. *Chronobiol Int*. 1998 Sep;15(5):425-45.
- h. Mrosovsky N. Locomotor activity and non-photic influences on circadian clocks. *Biol Rev Camb Philos Soc*. 1996 Aug;71(3):343-72.
- i. Mrosovsky N. A non-photic gateway to the circadian clock of hamsters. *Ciba Found Symp*. 1995;183:154-67; discussion 167-74.

2. An exhaustive review of the literature cannot be expected to accompany every journal article, but this study is about adenosine and SCN circadian clock function, and the failure to cite directly relevant literature is not acceptable. Studies by Watanabe et al (1996), Elliott et al (2001), Antle et al (2001) and Sigworth and Rea (2003) established that adenosine A1 agonists attenuate SCN responses to light. The Antle study further showed that in Syrian hamsters, an adenosine A1 agonist administered in the middle of the subjective day could induce phase shifts that mimic non-

photic manipulations, such as arousal stimulated by exercise (Mrosovsky, 1995) or by gentle handling, i.e., sleep deprivation without exercise (Antle and Mistlberger, 2000). Caffeine, an A1/A2a antagonist, did not induce phase shifts in the subjective day, but did attenuate phase shifts to wheel running evoked in the subjective day by confinement to a novel wheel (Antle et al 2001). Other studies have established that sleep deprivation can attenuate circadian phase shifting responses to light (e.g., Mistlberger et al, 1997), a finding consistent with studies showing that photic and nonphotic inputs to the SCN pacemaker are mutually antagonistic (e.g., studies by Mrosovsky and others). A different profile of effects may obtain in diurnal species, but the available evidence on that is limited (Jha et al, 2017; Burke et al, 2015, cited in the manuscript). In any case, the conclusion from these studies is that adenosine signaling within the SCN pacemaker contributes to effects of sleep-wake manipulations on circadian phase. This is very much consistent with the conclusions of the current study that "the biological relevance of adenosine signalling to circadian rhythms is to incorporate sleep history into the clockwork".

- a. Watanabe A, Moriya T, Nisikawa Y, Araki T, Hamada T, Shibata S, Watanabe S. Adenosine A1-receptor agonist attenuates the light-induced phase shifts and fos expression in vivo and optic nerve stimulation-evoked field potentials in the suprachiasmatic nucleus in vitro. *Brain Res.* 1996 Nov 18;740(1-2):329-36. doi:
- b. Elliott KJ, Todd Weber E, Rea MA. Adenosine A1 receptors regulate the response of the hamster circadian clock to light. *Eur J Pharmacol.* 2001 Feb 23;414(1):45-53. doi: 10.1016/s0014-2999(01)00786-5.
- c. Antle MC, Steen NM, Mistlberger RE. Adenosine and caffeine modulate circadian rhythms in the Syrian hamster. *Neuroreport.* 2001 Sep 17;12(13):2901-5. doi:
- d. Hallworth R, Cato M, Colbert C, Rea MA. Presynaptic adenosine A1 receptors regulate retinohypothalamic neurotransmission in the hamster suprachiasmatic nucleus. *J Neurobiol.* 2002;52(3):230-240. doi:10.1002/neu.10080
- e. Sigworth LA, Rea MA. Adenosine A1 receptors regulate the response of the mouse circadian clock to light. *Brain Res.* 2003 Jan 17;960(1-2):246-51. doi:
- f. Antle MC, Mistlberger RE. Circadian clock resetting by sleep deprivation without exercise in the Syrian hamster. *J Neurosci.* 2000 Dec 15;20(24):9326-32. doi:
- g. Mistlberger RE, Landry GJ, Marchant EG. Sleep deprivation can attenuate light-induced phase shifts of circadian rhythms in hamsters. *Neurosci Lett.* 1997 Nov 28;238(1-2):5-8. doi: 10.1016/s0304-3940(97)00815-x.
- h. van Diepen HC, Lucassen EA, Yassenkov R, et al. Caffeine increases light responsiveness of the mouse circadian pacemaker. *Eur J Neurosci.* 2014;40(10):3504-3511.

The authors are aware of at least some of this literature. The authors state in their response to the reviewers that "Our work has defined the mechanisms and the physiological relevance whereby adenosine signalling interacts with light for circadian regulation, moving the field forward by turning speculation into an evidence-based understanding." I agree that the studies reported here move the field forward, but the previously published evidence that adenosine modulates SCN phase and response to light is not speculation. A link between adenosine and behavioral effects on the clock was also not mere speculation, given that caffeine was shown to attenuate circadian phase shifts to exercise (3h awake) without reducing the amount of stimulated activity. These prior findings should be referenced, with appropriate attributions.

The authors claim that "we make the strong point that this literature is fragmentary with regard to circadian regulation and can only be placed into a biological context in the light of the novel mechanistic work we present in our manuscript. " The empirical findings that I cite above represent a coherent set of facts with biological/behavioural relevance. The authors interpretation of the literature appears disingenuous.

The authors state in their response that "We return to this point at the end of the discussion, where we detail how sleep/wake status via levels of adenosine, acts to alter light entrainment at dawn and dusk to achieve entrainment for both nocturnal and diurnal species." This is an example

of speculation. Intriguing, but you haven't shown that endogenous variations in adenosine are required to achieve entrainment. Furthermore, there are many exceptions to Ashoff's rules, particularly as these pertain to diurnal species.

3. Results - Line 150 - The authors state that "As predicted from the in vitro findings, both CGS and JNJ lengthened period in ex vivo cultured Per2::Luc SCN slices (Fig. 2c) and adenosine itself reduced period." However, line 88 states that, "in U2OS cells (in vitro findings), we observed that adenosine led to a concentration dependent period-lengthening". Observing that adenosine lengthens period in U2OS cells is not a rational basis for predicting that adenosine would shorten the circadian period of the SCN clock.

4. Results, line 169 - The authors state that " As adenosine receptor antagonists activate the same signalling pathways downstream from light within the SCN, we hypothesised that these antagonists will induce phase-shifts of the circadian clock following administration at specific times.... [the A2a/A1 antagonist] JNJ administered intraperitoneally six hours after light onset (zeitgeber time 6; ZT6) elicited large phase advances in circadian wheel-running behaviour" . If adenosine antagonists activate the same signaling pathways in SCN neurons as does light (input from the retina), then why would you test the drug at ZT6 (when light is not effective), and report a large phase advance (a non-photic type phase), without then noting and attempting to explain the evident conundrum? Based on what we know from prior studies, we would predict that the drug alone at ZT6 would not induce a shift, but might attenuate phase shifts induced by behavioural procedures that prevent sleep (e.g., running in a novel wheel, or sleep deprivation by gentle handling). Indeed, that is the result found by Antle et al 2001, using another A2a/A1 antagonist (caffeine)

5. Results, line 226 - The authors state that "Our findings are consistent with a previous study showing that there is decreased Per2 expression within the SCN following induced activity during the sleep phase³⁶, but differ from another report suggesting that there are no changes in Per2 within the SCN after sleep deprivation³⁷. We suggest that this discrepancy arises from the different methodological approaches used." It would be useful here to identify the methodological differences that might be at play (this does not need a long discussion, just something brief for the benefit of readers not intimate with this literature).

NCOMMS-19-29332A – Jagannath *et al.*

Please find our responses to the reviewers in blue, reviewers' comments produced verbatim.

REVIEWER COMMENTS

Reviewer #1 (Remarks to the Author):

1. >>The method of sleep deprivation used in this study was the one applied by the Franken group (the so-called 'gentle handling' method) and not by placing the animals in a novel wheel and increasing activity. In this context I would suggest that the authors do not simply write '... that this discrepancy arises from different methodological approaches used' on page 7 line 230-231, as this is cause for much confusion, but specify what part of the methods the authors think is important in causing the discrepancies.

The animals were allowed access to a running wheel throughout the sleep deprivation procedure, and the animals were seen to use the wheel. Further, the times at when the deprivation was conducted is different, therefore the text has been amended to:

Line 248: both the first study and ours allowed access to a running wheel and were conducted at the same time (ZT6-12), however the latter⁵⁵ was conducted without a running wheel, ZT0-6.

2. >>Citation 43 (Daan and Pittendrigh 1976) is not about comparing phase response curves between diurnal and nocturnal mammals. What Daan and Pittendrigh show is that there is a large variability in phase response curves in nocturnal rodents and that there seems to be a relation with the corresponding free running period. However, when one looks at the two extreme species in this paper (Figure 3 of Daan and Pittendrigh 1976), the mouse (*Mus musculus*) and the golden (or Syrian) hamster (*Mesocricetus auratus*) it is clear that the suggestion made by the authors in the end of the discussion cannot be right. The mouse phase response curve indeed fits very well with the authors idea as it has a small phase advance part and a large phase delay part, which fits with the idea that adenosine accumulating during the active phase is reducing the light response in the phase advance part. However, for the hamster the phase response curve shows quite the opposite phase. Both are nocturnal mammals and therefore active in the dark phase, yet in the hamster the phase advance part at the end of the night is much larger than the phase delay part at the beginning of the night. So, the statement made by the authors in their response that the data presented in Daan and Pittendrigh 1976 is 'entirely consistent with our proposal' seems to me to be incorrect.

What can be seen in Figure 9 of the Daan 2000 paper is that the average phase response curves do not differ very much between diurnal (7 species) and nocturnal mammals (12 species). In addition, it also shows that the advance and delay part of the phase response curve within the diurnal and nocturnal group do not differ that much either. Both speak against the suggestion made by the authors in the final part of the discussion in the present manuscript. I therefore urge the authors to remove this part of the discussion as it is not supported by data in the literature.

Because reviewer 2 had requested an additional figure explaining this discussion, we have addressed all reviewers' comments by discussing only mice. We then write that whilst these observations may not hold true all species, suggesting there are more factors at play here than just adenosine, we speculate that the reverse would be true in humans, as we sleep during the night. We have altered Suppl. Fig. 7 to reflect this.

Line 288: Under conditions of sleep deprivation, corresponding to a high adenosinergic tone, adenosine attenuates light-induced phase shifts. This observation is interesting in the context of entrainment in nocturnal and diurnal species. In all species studied, there is a phase response curve (PRC) to light such that light delays the circadian clock at dusk and advances the clock at dawn⁴³. In this manner, light exposure at dawn and dusk ensures that circadian rhythms are appropriately aligned/entrained to the solar day on a daily basis. ~~With some exceptions, diurnal species Mice have circadian rhythms that are significantly longer shorter than 24h, and so will show a daily phase delay advance in behavioural rhythms. By contrast, the clocks of nocturnal species are shorter (<24h) and phase advance. As a result, diurnal species mice require a daily phase advance delay to ensure that the clock is entrained to the 24h day and nocturnal species a phase delay. Without such advancing delaying cues, the endogenous delaying sleep/wake cycle of a diurnal animal mice will drift later and later earlier in time. Significantly, diurnal animals such as ourselves will mice would have high levels of adenosine at dusk dawn (following a day night of activity), whilst adenosine levels will be low at dawn dusk (following a night day of sleep). Because adenosine acts to inhibit the effects of light on the circadian clock, dusk light will induce only small large phase delays, whilst dawn light will induce large small phase advances. The net result will be a daily phase advance delay, which is precisely what is required to correct the delaying clock of a diurnal species to entrain. Whilst these observations may not hold true in all species, suggesting there are more factors at play here than just adenosine, we speculate that the reverse would be true in humans, as we sleep during the night and have longer clocks (>24h) The reverse will be true for a nocturnal species, which have shorter clocks (<24h) and require a delaying phase shifts at dusk to entrain. Sleep during the day will lower adenosine, making the clock more sensitive to light at dusk and delaying phase shifts, whilst activity at night will elevate adenosine making the clock less sensitive to light at dawn and advancing phase shifts. In this manner, light and sleep/wake history could interact to achieve optimal sleep/wake timing (Suppl. Fig. 7).~~

Reviewer #2 (Remarks to the Author):

This is a revised version of a manuscript titled: "Adenosine integrates light and sleep signalling for the regulation of circadian timing".

I raised a number of concerns with the previous version and feel that the authors addressed these concerns.

In broad strokes, the authors use pharmacological and genetic approaches to provide evidence that demonstrate that adenosine acts upon the circadian clockwork via adenosine A1/A2A receptor signalling through the activation of the Ca²⁺-ERK-AP-1 and CREB/CRTC1-CRE pathways to regulate the clock genes Per1 and Per2. They show that these signalling pathways converge upon and inhibit the same pathways activated by light. Thus, circadian entrainment by light can be modulated by sleep history and provides a nice concrete example of how process S and C can interact.

This is exciting and should generate broad interest. I continue to be impressed with the combination of data from cell culture, SCN biochemistry, and behaviour.

From my point of view, the problem that the other reviewers and I struggle with is that there are so many details that the main point gets lost. But the writing is generally clear and help

the reader follow the linear flow of the story. The manuscript could likely benefit from even more pruning of the figures. But I do recognize that a lot of work went into this study.

Our study reflects the complexity of adenosine signalling. But rather than oversimplify the work presented we would prefer to provide the reader with a full account of the data-set and so not remove any of the Figures. We would of course be happy to take a further look if the editor felt that this was essential.

I have a few additional points (minor).

In Fig1d, are negative controls available? Similarly, for Fig.1i, are positive controls available? The activation and inhibition of pCREB would be more striking with the controls.

1d and 1i are part of the same western blot, where DMSO serves as the negative control and Forskolin as the positive. The entire western blot has been provided for reference in the supplementary material and the legend has been altered to reflect this.

For Fig1e, j were two way ANOVA used to assess which gene expression values are significantly different?

This information is now included in the figure legend.

For Fig1o, can the authors provide more explanation of the siRNA experiment. I was struggling with the visual display of this experiment even if the point is clear.

We have included the following text, line 135. To define the signalling elements between adenosine receptors and AP-1, we silenced a series of members of the MAPK-ERK signalling pathway that have been shown to activate AP-1¹⁸ in U2OS cells, and then tested the efficacy of CGS at lengthening period using a concentration response curve (Fig. 1n). We then calculated EC50 of CGS at period lengthening under each knockdown condition from the curves (Fig. 1o) and found that silencing ERK1 abolished the period-lengthening effect of CGS (Fig. 1n-o).

In Fig2a, it looks like the receptor is not expressed in the night. Can the authors provide more explanation for how they seen regulation of the light during the night? Is the receptors half-life sufficient that the A₁ made in the day would be present at night?

The receptor is expressed at night and detected by PCR; but just at a third of the level during the day, these data are included now in the legend (line 372).

In Fig2d the authors are showing the drugs increase pCREB. Can the authors help the reader understand how this fits with Fig1i which apparently does not show pCREB activation?

We have added the following text, also addressing a concern of Reviewer 4.

Line 153: ~~As predicted from the in vitro findings, both CGS and JNJ lengthened period in ex vivo cultured Per2::Luc SCN slices (Fig. 2c) and adenosine itself reduced period presumably due to the prevalence of Gi coupled A₁ receptors. Also, and as anticipated due to the prevalence of A₁, CGS and JNJ increased ERK and CREB phosphorylation (Fig. 2d) and induced Per1/2 (Fig. 2e). We speculate that this is likely due to the different receptor distribution within the SCN, with inhibitory A₁ receptors predominating. Here, the dominant effect of the antagonist would be inhibition of the Gi-PKA system, which would lead to an increase in cAMP and thus CREB~~

phosphorylation. In contrast, the A_{2B} receptors present in U2OS cells are G_s and G_q coupled, therefore their antagonism would not lead to CREB phosphorylation. However, adenosine receptor signalling is known to be very complex, with adenosine receptors with adenosine receptors forming heterodimers, modulating the release of other signalling messengers and neurotransmitters, and thus engaging different pathways depending on receptor distribution²⁶. Therefore, what is provided above is a proposed mechanism based on the available data.

26 *Klinger, M., Freissmuth, M. & Nanoff, C. Adenosine receptors: G protein-mediated signalling and the role of accessory proteins. Cell Signal 14, 99-108, doi:10.1016/s0898-6568(01)00235-2 (2002).*

Reviewer #4 (Remarks to the Author):

This is an interesting set of experiments, pursuing a topic that was investigated by several groups about 20 years ago. The study confirms findings from prior in vivo behavioral pharmacology studies and extends these to define intracellular signaling pathways. I agree with previous reviewer comments that the results are confusing in some respects (see below). There are also issues with scholarship, noted in the first round of reviews, but not addressed in this revision. The authors fail to acknowledge prior work, although they do appear to be aware of it, based on the response to the reviews.

Introduction

1. page 2, line 56 - The authors state that "By contrast, the SCN is almost exclusively entrained by light and is unresponsive to such stimuli³". This is factually incorrect; the SCN is neither exclusively entrained by light, nor is it unresponsive to non-photic stimuli. There is copious evidence that the SCN pacemaker in sighted and blind animals can be shifted and entrained by non-photic stimuli, including scheduled activity and feeding. Sensitivity may vary by species and strain (SCN controlled free-running rhythms in some mouse strains entrain to feeding schedules; in others this is less likely, and in some species even less so; species differences in phase shifts to scheduled exercise are also known), but these effects are not trivial, can compete with light, may be more important in natural environments (more studies needed), and should not be dismissed. There are many review papers that can be cited here [below is a selection from a quick pubmed search]. This needs to be corrected so that readers new or peripheral to the field are properly informed.

- a. Webb IC, Antle MC, Mistlberger RE. Regulation of circadian rhythms in mammals by behavioral arousal. *Behav Neurosci.* 2014 Jun;128(3):304-25.
- b. Challet E. Minireview: Entrainment of the suprachiasmatic clockwork in diurnal and nocturnal mammals. *Endocrinology.* 2007 Dec;148(12):5648-55.
- c. Mistlberger RE, Skene DJ. Nonphotic entrainment in humans? *J Biol Rhythms.* 2005 Aug;20(4):339-52.
- d. Yannielli P, Harrington ME. Let there be "more" light: enhancement of light actions on the circadian system through non-photic pathways. *Prog Neurobiol.* 2004 Sep;74(1):59-76.
- e. Mistlberger RE, Skene DJ. Social influences on mammalian circadian rhythms: animal and human studies. *Biol Rev Camb Philos Soc.* 2004 Aug;79(3):533-56.
- f. Challet E, Pévet P. Interactions between photic and nonphotic stimuli to synchronize the master circadian clock in mammals. *Front Biosci.* 2003 May 1;8:s246-57.
- g. Hastings MH, Duffield GE, Smith EJ, Maywood ES, Ebling FJ. Entrainment of the circadian system of mammals by nonphotic cues. *Chronobiol Int.* 1998 Sep;15(5):425-45.
- h. Mrosovsky N. Locomotor activity and non-photic influences on circadian clocks. *Biol Rev Camb Philos Soc.* 1996 Aug;71(3):343-72.
- i. Mrosovsky N. A non-photic gateway to the circadian clock of hamsters. *Ciba Found Symp.* 1995;183:154-67; discussion 167-74.

There is substantial evidence that the SCN is primarily entrained by light. However we accept the referees point that other stimuli can entrain the SCN, and the text has been modified to accommodate this point, and several of the suggested papers above have been cited.

Line 56: By contrast, the SCN is primarily entrained by light. For example, the SCN is largely unresponsive to glucocorticoids³, although there is evidence that non-photic stimuli contribute to SCN entrainment⁴⁻⁸. However, the SCN is sensitive to caffeine⁹⁻¹¹...

3 Balsalobre, A. et al. Resetting of circadian time in peripheral tissues by glucocorticoid signaling. Science 289, 2344-2347, doi:8856 [pii] (2000).

4 Welsh, D. K., Takahashi, J. S. & Kay, S. A. Suprachiasmatic nucleus: cell autonomy and network properties. Annu Rev Physiol 72, 551-577, doi:10.1146/annurev-physiol-021909-135919 (2010).

5 Golombek, D. A. & Rosenstein, R. E. Physiology of circadian entrainment. Physiol Rev 90, 1063-1102, doi:10.1152/physrev.00009.2009 (2010).

6 Webb, I. C., Antle, M. C. & Mistlberger, R. E. Regulation of circadian rhythms in mammals by behavioral arousal. Behav Neurosci 128, 304-325, doi:10.1037/a0035885 (2014).

7 Challet, E. & Pevet, P. Interactions between photic and nonphotic stimuli to synchronize the master circadian clock in mammals. Front Biosci 8, s246-257, doi:10.2741/1039 (2003).

8 Hastings, M. H., Duffield, G. E., Smith, E. J., Maywood, E. S. & Ebling, F. J. Entrainment of the circadian system of mammals by nonphotic cues. Chronobiol Int 15, 425-445 (1998).

9 Oike, H., Kobori, M., Suzuki, T. & Ishida, N. Caffeine lengthens circadian rhythms in mice. Biochemical and biophysical research communications 410, 654-658, doi:10.1016/j.bbrc.2011.06.049 (2011).

10 Burke, T. M. et al. Effects of caffeine on the human circadian clock in vivo and in vitro. Sci Transl Med 7, 305ra146, doi:10.1126/scitranslmed.aac5125 (2015).

11 Antle, M. C., Steen, N. M. & Mistlberger, R. E. Adenosine and caffeine modulate circadian rhythms in the Syrian hamster. Neuroreport 12, 2901-2905, doi:10.1097/00001756-200109170-00029 (2001).

2. An exhaustive review of the literature cannot be expected to accompany every journal article, but this study is about adenosine and SCN circadian clock function, and the failure to cite directly relevant literature is not acceptable. Studies by Watanabe et al (1996), Elliott et al (2001), Antle et al (2001) and Sigworth and Rea (2003) established that adenosine A1 agonists attenuate SCN responses to light. The Antle study further showed that in Syrian hamsters, an adenosine A1 agonist administered in the middle of the subjective day could induce phase shifts that mimic non-photic manipulations, such as arousal stimulated by exercise (Mrosovsky, 1995) or by gentle handling, i.e., sleep deprivation without exercise (Antle and Mistlberger, 2000). Caffeine, an A1/A2a antagonist, did not induce phase shifts in the subjective day, but did attenuate phase shifts to wheel running evoked in the subjective day by confinement to a novel wheel (Antle et al 2001). Other studies have established that sleep deprivation can attenuate circadian phase shifting responses to light (e.g., Mistlberger et al, 1997), a finding consistent with studies showing that photic and nonphotic inputs to the SCN pacemaker are mutually antagonistic (e.g., studies by Mrosovsky and others). A different profile of effects may obtain in diurnal species, but the

available evidence on that is limited (Jha et al, 2017; Burke et al, 2015, cited in the manuscript). In any case, the conclusion from these studies is that adenosine signaling within the SCN pacemaker contributes to effects of sleep-wake manipulations on circadian phase. This is very much consistent with the conclusions of the current study that "the biological relevance of adenosine signalling to circadian rhythms is to incorporate sleep history into the clockwork".

- a. Watanabe A, Moriya T, Nisikawa Y, Araki T, Hamada T, Shibata S, Watanabe S. Adenosine A1-receptor agonist attenuates the light-induced phase shifts and fos expression in vivo and optic nerve stimulation-evoked field potentials in the suprachiasmatic nucleus in vitro. *Brain Res.* 1996 Nov 18;740(1-2):329-36. doi:
- b. Elliott KJ, Todd Weber E, Rea MA. Adenosine A1 receptors regulate the response of the hamster circadian clock to light. *Eur J Pharmacol.* 2001 Feb 23;414(1):45-53. doi: 10.1016/s0014-2999(01)00786-5.
- c. Antle MC, Steen NM, Mistlberger RE. Adenosine and caffeine modulate circadian rhythms in the Syrian hamster. *Neuroreport.* 2001 Sep 17;12(13):2901-5. doi:
- d. Hallworth R, Cato M, Colbert C, Rea MA. Presynaptic adenosine A1 receptors regulate retinohypothalamic neurotransmission in the hamster suprachiasmatic nucleus. *J Neurobiol.* 2002;52(3):230-240. doi:10.1002/neu.10080
- e. Sigworth LA, Rea MA. Adenosine A1 receptors regulate the response of the mouse circadian clock to light. *Brain Res.* 2003 Jan 17;960(1-2):246-51. doi:
- f. Antle MC, Mistlberger RE. Circadian clock resetting by sleep deprivation without exercise in the Syrian hamster. *J Neurosci.* 2000 Dec 15;20(24):9326-32. doi:
- g. Mistlberger RE, Landry GJ, Marchant EG. Sleep deprivation can attenuate light-induced phase shifts of circadian rhythms in hamsters. *Neurosci Lett.* 1997 Nov 28;238(1-2):5-8. doi: 10.1016/s0304-3940(97)00815-x.
- h. van Diepen HC, Lucassen EA, Yassenkov R, et al. Caffeine increases light responsiveness of the mouse circadian pacemaker. *Eur J Neurosci.* 2014;40(10):3504-3511.

We have included the following:

Line 234: It has been shown previously that sleep deprivation attenuates phase-shifts to light, and also that A1 receptors attenuate light signalling^{16,49-53}. Indeed, we confirm that sleep deprivation...

- 49 *Watanabe, A. et al. Adenosine A1-receptor agonist attenuates the light-induced phase shifts and fos expression in vivo and optic nerve stimulation-evoked field potentials in the suprachiasmatic nucleus in vitro. Brain research 740, 329-336, doi:10.1016/s0006-8993(96)00881-5 (1996).*
- 50 *Elliott, K. J., Todd Weber, E. & Rea, M. A. Adenosine A1 receptors regulate the response of the hamster circadian clock to light. Eur J Pharmacol 414, 45-53, doi:10.1016/s0014-2999(01)00786-5 (2001).*
- 51 *Hallworth, R., Cato, M., Colbert, C. & Rea, M. A. Presynaptic adenosine A1 receptors regulate retinohypothalamic neurotransmission in the hamster suprachiasmatic nucleus. J Neurobiol 52, 230-240, doi:10.1002/neu.10080 (2002).*
- 52 *Sigworth, L. A. & Rea, M. A. Adenosine A1 receptors regulate the response of the mouse circadian clock to light. Brain research 960, 246-251 (2003).*

53 *Mistlberger, R. E., Landry, G. J. & Marchant, E. G. Sleep deprivation can attenuate light-induced phase shifts of circadian rhythms in hamsters. Neurosci Lett 238, 5-8 (1997).*

In addition, van Diepen et al. are already referred to (line 67) and Antle et al. included (line 56), see above.

The authors are aware of at least some of this literature. The authors state in their response to the reviewers that "Our work has defined the mechanisms and the physiological relevance whereby adenosine signalling interacts with light for circadian regulation, moving the field forward by turning speculation into an evidence-based understanding." I agree that the studies reported here move the field forward, but the previously published evidence that adenosine modulates SCN phase and response to light is not speculation. A link between adenosine and behavioral effects on the clock was also not mere speculation, given that caffeine was shown to attenuate circadian phase shifts to exercise (3h awake) without reducing the amount of stimulated activity. These prior findings should be referenced, with appropriate attributions.

The authors claim that "we make the strong point that this literature is fragmentary with regard to circadian regulation and can only be placed into a biological context in the light of the novel mechanistic work we present in our manuscript. " The empirical findings that I cite above represent a coherent set of facts with biological/behavioural relevance. The authors interpretation of the literature appears disingenuous.

It was not our intention to be disingenuous. But as the referee states he/she has provided "a coherent set of facts" – we would make the point that these are stand-alone facts – they are given much greater meaning, and indeed understanding, in the context of the new data we present. The significance of these findings can now placed into a functional context.

The authors state in their response that "We return to this point at the end of the discussion, where we detail how sleep/wake status via levels of adenosine, acts to alter light entrainment at dawn and dusk to achieve entrainment for both nocturnal and diurnal species." This is an example of speculation. Intriguing, but you haven't shown that endogenous variations in adenosine are required to achieve entrainment. Furthermore, there are many exceptions to Ashoff's rules, particularly as these pertain to diurnal species.

This point has been responded to and accommodated in the response to Reviewer 1, please see line 284.

3. Results - Line 150 - The authors state that "As predicted from the in vitro findings, both CGS and JNJ lengthened period in ex vivo cultured Per2::Luc SCN slices (Fig. 2c) and adenosine itself reduced period." However, line 88 states that, "in U2OS cells (in vitro findings), we observed that adenosine led to a concentration dependent period-lengthening". Observing that adenosine lengthens period in U2OS cells is not a rational basis for predicting that adenosine would shorten the circadian period of the SCN clock.

We speculated the above on the basis of receptor distribution (A_1 in the SCN but A_{2B} in U2OS). We have taken on board the recommendations of this reviewer and Reviewer2, please see our response above that begins as follows:

Line 150: ~~As predicted from the in vitro findings, both CGS and JNJ lengthened period in ex vivo cultured Per2::Luc SCN slices (Fig. 2c)...~~

4. Results, line 169 - The authors state that " As adenosine receptor antagonists activate the same signalling pathways downstream from light within the SCN, we hypothesised that these antagonists will induce phase-shifts of the circadian clock following administration at specific times.... [the A_{2a}/A₁ antagonist] JNJ administered intraperitoneally six hours after light onset (zeitgeber time 6; ZT6) elicited large phase advances in circadian wheel-running behaviour". If adenosine antagonists activate the same signaling pathways in SCN neurons as does light (input from the retina), then why would you test the drug at ZT6 (when light is not effective), and report a large phase advance (a non-photoc type phase), without then noting and attempting to explain the evident conundrum? Based on what we know from prior studies, we would predict that the drug alone at ZT6 would not induce a shift, but might attenuate phase shifts induced by behavioural procedures that prevent sleep (e.g., running in a novel wheel, or sleep deprivation by gentle handling). Indeed, that is the result found by Antle et al 2001, using another A_{2a}/A₁ antagonist (caffeine).

We conducted a phase-response curve with Istradefylline (KW6002) administration and found that the peak advance zone was around ZT6-8 (Supplementary Fig. 4h). We considered these data together with several other reports that non-photoc cues including activity, melatonin and cAMP itself cause advances at ZT6. Further, caffeine has been shown to cause phase advances in peripheral tissues at this time (Narishige et al., 2014).

We agree with this reviewer that the phase advancing effect we observe of A_{2a}/A₁ antagonists is not consistent with light, although the underlying pathways (cAMP or ERK/AP1) are those that are also downstream of light. Therefore, we have modified the text to:

Line 181: As adenosine receptor antagonists activate the signalling pathways downstream from light within the SCN, we hypothesised that these antagonists will induce phase-shifts of the circadian clock following administration at specific times during the night. In addition, several non-photoc cues, including cAMP itself induce phase advances of the clock in vitro and in vivo during the day³²⁻⁴³ and a few enhance re-entrainment to a shifted light dark cycle³⁵⁻³⁷. Therefore, we tested the effects of JNJ at times that cause maximum delay in response to light (CT16), but also when non-photoc cues would elicit a large advance (ZT6), and in a re-entrainment paradigm.

32 Lewy, A. J. et al. The human phase response curve (PRC) to melatonin is about 12 hours out of phase with the PRC to light. *Chronobiol Int* 15, 71-83, doi:10.3109/07420529808998671 (1998).

33 Reeb, S. G. & Mrosovsky, N. Effects of induced wheel running on the circadian activity rhythms of Syrian hamsters: entrainment and phase response curve. *J Biol Rhythms* 4, 39-48, doi:10.1177/074873048900400103 (1989).

34 Biello, S. M. & Mrosovsky, N. Phase response curves to neuropeptide Y in wildtype and tau mutant hamsters. *J Biol Rhythms* 11, 27-34, doi:10.1177/074873049601100103 (1996).

35 Agostino, P. V., Plano, S. A. & Golombek, D. A. Sildenafil accelerates reentrainment of circadian rhythms after advancing light schedules. *Proc Natl Acad Sci U S A* 104, 9834-9839, doi:10.1073/pnas.0703388104 (2007).

- 36 *Pilorz, V. et al. A novel mechanism controlling resetting speed of the circadian clock to environmental stimuli. Curr Biol 24, 766-773, doi:10.1016/j.cub.2014.02.027 (2014).*
- 37 *Prosser, R. A. & Gillette, M. U. The mammalian circadian clock in the suprachiasmatic nuclei is reset in vitro by cAMP. J Neurosci 9, 1073-1081 (1989).*

5. Results, line 226 - The authors state that "Our findings are consistent with a previous study showing that there is decreased Per2 expression within the SCN following induced activity during the sleep phase³⁶, but differ from another report suggesting that there are no changes in Per2 within the SCN after sleep deprivation³⁷. We suggest that this discrepancy arises from the different methodological approaches used." It would be useful here to identify the methodological differences that might be at play (this does not need a long discussion, just something brief for the benefit of readers not intimate with this literature).

This is now expanded as recommended also by Reviewer 1.

Line 231: ...both the first study and ours allowed access to a running wheel and were conducted at the same time (ZT6-12), however the latter was conducted without a running wheel, ZT0-6.

Reviewers' Comments:

Reviewer #1:

Remarks to the Author:

Response: The referee is primarily concerned that our results differ from those of the Franken group, which report that sleep deprivation does not alter clock gene expression in the SCN. However, the referee seems to have overlooked some important, indeed fundamental, points. Critically, the two studies were undertaken at different times. Our sleep deprivation was undertaken at ZT 6-12, whereas the Franken group (Curie et al. 2015) was undertaken at ZT 0- 6. The Franken group makes the point that sleep deprivation at different times has opposite effects upon clock gene expression (Curie et al. 2013). Indeed, our unpublished data shows that sleep deprivation between ZT 0-6 similarly produces no change in clock gene expression within the SCN. Thus, in this regard, our findings are directly comparable with those of Curie et. al..

Reviewer 1 states that it is inappropriate to compare our finding with those of Maywood et. al. We would point out that: 1) Our sleep deprivation studies, and those of Maywood, were undertaken at the same time; 2) The readout for our studies and those of Maywood used measurements of mRNA – rather than protein (Curie et al. 2015); 3) Both the Maywood study and our study used a running wheel. Collectively, therefore, it is no surprise that our results agreed with Maywood et al. and we feel fully justified in making this comparison. Beyond our study, and the papers of Maywood et al. and Curie et al. there are no others reports of the effects of sleep deprivation on clock gene expression in the mouse SCN, and we have fully discussed our results within the context of these two studies.

The method of sleep deprivation used in this study was the one applied by the Franken group (the so-called 'gentle handling' method) and not by placing the animals in a novel wheel and increasing activity (Maywood). In this context I would suggest that the authors do not simply write '... that this discrepancy arises from different methodological approaches used' on page 7 line 230-231, as this is cause for much confusion, but specify what part of the methods the authors think is important in causing the discrepancies.

We object strongly to the statement, "the authors make up a story which is not substantiated with references". This is entirely incorrect and appears designed to misrepresent our work. We cite Serge Daan's studies, which are entirely consistent with our proposal. The collection of four back-to-back papers by Daan and Pittendrigh, *J. Comp. Physiol.* 106, 223-331 (1976) have over two thousand citations, and collectively highlight the differences in the advance and delay portions of the phase response curves (PRCs) between nocturnal and diurnal species. Whilst a few exceptions exist, species with a shorter free-running period (generally nocturnal) show larger delays than advances, whilst species with a longer free-running period (generally diurnal) show larger advances compared to delays. This is covered in the abstract of the paper we cite (reference 43), and indeed we discussed our findings with Prof. Daan before his untimely death. We are very surprised this Reviewer has ignored these key papers and has referred instead to the transcript of a conference lecture by Serge Daan, where he discusses the "dead zone" of the PRC and not the size of advances and delays in diurnal vs nocturnal species.

Citation 43 (Daan and Pittendrigh 1976) is not about comparing phase response curves between diurnal and nocturnal mammals. The title reads: " A functional analysis of circadian pacemakers in nocturnal rodents. II The paper is only showing (a very thorough analysis of) nocturnal rodents and their phase response curves. What Daan and Pittendrigh show is that there is a large variability in phase response curves in nocturnal rodents and that there seems to be a relation with the corresponding free running period. However, when one looks at the two extreme species in this paper (Figure 3 of Daan and Pittendrigh 1976), the mouse (*Mus musculus*) and the golden (or Syrian) hamster (*Mesocricetus auratus*) it is clear that the suggestion made by the authors in the end of the discussion cannot be right. The mouse phase response curve indeed fits very well with the authors idea. However, for the hamster the phase response curve shows the opposite. Both are nocturnal mammals and therefore active in the dark phase, yet in the hamster the phase

advance part at the end of the night is much larger than the phase delay part at the beginning of the night. So, the statement made by the authors that the data presented in Daan and Pittendrigh 1976 is 'entirely consistent with our proposal' seems to me to be incorrect.

What can be seen in Daan 2000, Figure 9 is that the average phase response curves do not differ very much between diurnal (7 species) and nocturnal mammals (12 species). In addition, it also shows that the advance and delay part of the phase response curve within the diurnal and nocturnal group do not differ that much. Both speak against the suggestion made by the authors in the final part of the discussion.

Reviewer #2:

Remarks to the Author:

This is a revised version (#2) of a manuscript titled: "Adenosine integrates light and sleep signalling for the regulation of circadian timing".

I raised several concerns with the previous version and feel that the authors addressed these concerns.

The findings are generally exciting and should generate broad interest. I continue to be impressed with the combination of data from cell culture, SCN biochemistry, and behaviour.

Reviewer #4:

Remarks to the Author:

The revisions are acceptable. No further comments.

NCOMMS-19-29332C – Jagannath *et al.*

Please find our responses to the reviewers in blue, reviewers' comments produced verbatim.

REVIEWER COMMENTS

Reviewer #1 (Remarks to the Author):
[REDACTED]

Reviewer #2 (Remarks to the Author):

This is a revised version (#2) of a manuscript titled: "Adenosine integrates light and sleep signalling for the regulation of circadian timing".

I raised several concerns with the previous version and feel that the authors addressed these concerns.

The findings are generally exciting and should generate broad interest. I continue to be impressed with the combination of data from cell culture, SCN biochemistry, and behaviour.

Many thanks, we appreciate the interest and constructive feedback.

Reviewer #4 (Remarks to the Author):

The revisions are acceptable. No further comments.

Many thanks for the useful feedback which has strengthened the manuscript.